# Membrane-induced 2D phase separation of the focal adhesion protein talin

Thomas Litschel [1,4,6] ✉, Charlotte F. Kelley [1,6], Xiaohang Cheng [2], Leon Babl [1], Naoko Mizuno [3,5], Lindsay B. Case [2] & Petra Schwille [1] ✉

Focal adhesions form liquid-like assemblies around activated integrin receptors at the plasma membrane. How they achieve their flexible properties is not well understood. Here, we use recombinant focal adhesion proteins to reconstitute the core structural machinery in vitro. We observe liquid-liquid phase separation of the core focal adhesion proteins talin and vinculin for a spectrum of conditions and interaction partners. Intriguingly, we show that binding to PI(4,5)P$_2$-containing membranes triggers phase separation of these proteins on the membrane surface, which in turn induces the enrichment of integrin in the clusters. We suggest a mechanism by which 2-dimensional biomolecular condensates assemble on membranes from soluble proteins in the cytoplasm: lipid-binding triggers protein activation and thus, liquid-liquid phase separation of these membrane-bound proteins. This could explain how early focal adhesions maintain a structured and force-resistant organization into the cytoplasm, while still being highly dynamic and able to quickly assemble and disassemble.

Focal adhesions connect the actin cytoskeleton to the extracellular matrix, thereby regulating cell shape, migration, differentiation, and responses to extracellular stimuli[1]. Centered around activated integrin receptors in the plasma membrane, focal adhesions are highly dynamic assemblies made up of densely packed proteins[2,3]. These complexes can form and disassemble rapidly in response to cellular cues, or form stable, long-lasting connections between the extracellular matrix and cellular actomyosin networks. Focal adhesion proteins interact mainly through weak, often multivalent, interactions and can rapidly exchange between focal adhesions and the cytoplasm[4–6]. Individual focal adhesions grow and mature over time and can fuse with each other[7–9]. Mathematical models have concluded that protein concentrations required for the fast, dynamic assembly of focal adhesions far exceed average concentrations on the cell membrane and in the cytoplasm, and that an additional level of organization is necessary to result in local enrichment of proteins[10]. Intriguingly, all these defining characteristics of focal adhesions also apply to the group of protein assemblies referred to as biomolecular condensates, which form through liquid-liquid phase separation (LLPS). Thus, characterizing focal adhesions as phase-separated molecular assemblies could be the key to understanding their formation, dynamics, and role as cellular mechanosensitive signaling hubs.

Biomolecular condensates typically form through multivalent interactions from a few core components, referred to as scaffolds. Hallmarks of biomolecular condensates are their dynamic nature, rapid exchange of components, sensitivity to changes in cellular conditions, and ability to undergo fusion and fission events. Recent in vitro and cellular studies demonstrate that several focal adhesion adapter proteins undergo phase separation, and that phase separation may contribute to focal adhesion formation and function. Phase separation of the adapter proteins FAK, paxillin, and p130Cas promote kindlin-dependent integrin clustering on supported lipid bilayers and initial

[1]Department of Cellular and Molecular Biophysics, Max Planck Institute of Biochemistry, Martinsried, Germany. [2]Department of Biology, Massachusetts Institute of Technology, Cambridge, MA, USA. [3]Department of Structural Cell Biology, Max Planck Institute of Biochemistry, Martinsried, Germany. [4]Present address: John A. Paulson School of Engineering and Applied Sciences, Harvard University, Cambridge, MA, USA. [5]Present address: Laboratory of Structural Cell Biology, National Institutes of Health, Bethesda, MD, USA. [6]These authors contributed equally: Thomas Litschel, Charlotte F. Kelley.
✉e-mail: tlitschel@seas.harvard.edu; schwille@biochem.mpg.de

nascent focal adhesion assembly in cells[11–13]. The mechanosensitive adapter protein LIMD1 undergoes phase separation in vitro and in cells to form a condensed phase that recruits a specific subset of focal adhesion proteins. LIMD1 is recruited to focal adhesions under force, and perturbing LIMD1 phase separation alters focal adhesion dynamics, cellular mechanics, and durotaxis[14]. The focal adhesion protein Tensin1 undergoes phase separation during focal adhesion disassembly[15]. Thus, growing experimental evidence suggests that phase separation contributes to focal adhesion formation, maturation, and disassembly. However, mature focal adhesions in cells have distinct vertical layers of protein localization above the membrane[16]. How can phase separation of integrin receptors regulate protein organization 100 nm above the membrane surface? Identifying the underlying principles governing focal adhesion organization, composition, and dynamics is crucial for understanding the unique and varied roles these complexes play in cell adhesion.

Talin, a large (272 kDa) mechanosensitive structural-scaffolding protein (Fig. 1a), is an ideal candidate for regulating both integrin clustering and protein organization above the membrane. Talin connects integrin receptors to the actin cytoskeleton, and is essential for focal adhesion formation[17,18]. Roughly 100 nm long, talin has a highly polarized orientation within focal adhesions: its N-terminus localizes with integrin receptors at the membrane while its C-terminus localizes to sites of stress fiber attachment. Talin spans the entirety of focal adhesion complexes and is the primary determinant of the nanoscale organization within focal adhesions[19,20]. However, talin is a cytoplasmic protein, and typical phase separation of talin in the cytosol would not explain the highly ordered, layered structure of focal adhesions on membranes.

Biomolecular condensates have been shown to wet membranes[21–23] and their formation can be assisted by membranes through lowering the phase transition threshold[24–27], however these voluminous droplets still lack internal order. In fact, by definition, 3-dimensional condensates formed through LLPS are amorphous, anisotropic structures. In contrast to this, true membrane proteins have an intrinsic orientation, which, in many cases, is tied to their function. These kind of proteins have also been observed to phase separate, as mentioned for example in the case of integrin, and recently also receptor clustering within the plasma has been linked to transmembrane proteins undergoing liquid-liquid phase separation[28,29]. This results in the assembly of adhesion and signaling complexes in two dimensions (2D), with implications for down-stream signaling and the physical properties of receptors at the membrane surface. With the membrane as a structural platform, these condensates contain an intrinsic anisotropy and polarity, which can introduce a layer of organization 3-dimensional droplets do not allow for. Here, we explore whether proteins from solution, instead of condensing into 3-dimensional amorphous droplets, can condense into such 2-dimensional structures on the membrane. Bringing together the properties of biomolecular condensates and the polarity of membrane-bound talin results in a gain of function scenario that explains how focal adhesions can form highly organized, dynamic hubs made up of hundreds of proteins.

Due to its central role in focal adhesion formation and the organization of focal adhesion nanostructure, we characterized the ability of talin to form biomolecular condensates in vitro using purified recombinant proteins. We found that talin in solution undergoes phase separation to form 3D liquid condensates. In the cytoplasm, talin is highly regulated and is maintained in an autoinhibited, inactive conformation[20]. We therefore explored interactions between talin and other core components of focal adhesions suspected to release talin autoinhibition, namely vinculin, integrin and the phospholipid phosphatidylinositol(4,5)bisphosphate (PIP$_2$). We observe talin condensation for all combinations and thus conclude that with the release of its autoinhibition, talin is enabled to undergo liquid-liquid phase separation. Accordingly, surface interactions between talin and PIP$_2$-containing membranes result in the activation of membrane-bound talin and therefore leads to the formation of 2-dimensional talin condensates. Finally, we observed that this PIP$_2$-mediated talin clustering strengthens talin-membrane connections.

Our observations represent a case in which proteins from solution phase separate into 2-dimensional clusters instead of amorphous droplets, and thus form ordered, directional, molecular assemblies on the membrane. As such, the lipid-driven phase separation of talin could lend focal adhesions both the physical properties of molecular condensates and the directional molecular organization intrinsic to talin as a focal adhesion scaffold protein. This mechanism is made possible through the central role lipids play in the phase separation process, a regulatory mechanism likely applicable to other membrane-associated condensates.

## Results

### Talin phase-separates in solution upon vinculin binding

We first characterized the interactions between recombinant full-length talin and recombinant vinculin (Fig. 1a). Vinculin interacts directly with talin and actin, and is recruited to growing focal adhesions to regulate mechanotransduction. Both talin and vinculin need to be relieved of their autoinhibited conformation in order to interact (Fig. 1b)[20,30,31]. In order to focus on the activation of talin, we designed a deregulated vinculin (Vn$^{DR}$) by disrupting the two autoinhibitory interactions between the vinculin head and vinculin tail domains[31,32]. Without these two interactions, vinculin is able to interact with talin under low salt conditions (Fig. S1a). When mixed, full-length talin and deregulated vinculin form liquid-liquid phase separated droplets (Fig. 1c). Both autoinhibitory interactions within vinculin must be disrupted to observe this effect (Fig. S1b).

Vertebrates contain two talin genes, Tn1 and Tn2, which are 76% identical though not functionally redundant[33]. We compared the two talin isoforms, Tn1 and Tn2, mixed with Vn$^{DR}$ and used methyl cellulose as a crowding agent to imitate conditions in the cytoplasm. Under these conditions, Tn2 formed micrometer sized droplets, while Tn1 formed smaller droplets and produced a drastically lower volume of phase separated material (Fig. 1c), even at higher protein concentrations (Fig. S2a). It is possible that Tn1 requires further activation, possibly through an additional binding partner, to reach the level of phase separation observed with Tn2. We tested whether actin can play an activating role, but did not observe evidence of an increase in phase separation (Fig. S3). DeePhase, a predictor of protein phase behavior based on amino acid sequence, did not find talin1 any less likely to phase separate when compared to talin2[34]. Therefore, the different thresholds for phase separation observed for talin1 and talin2 are likely due to small differences in the strength of intramolecular interactions, regulating the tertiary structure of the autoinhibited protein.

We confirmed the liquid-like nature of the talin-vinculin droplets by observing the coalescence of multiple droplets and through fluorescence recovery after photobleaching (FRAP) measurements (Movie 1, Fig. 1d, e, Fig. S2b, c and Fig. S4). Recovery speed decreased over time for Tn2-Vn droplets, suggesting that these condensates mature and thereby become more viscous over time[35]. The volume of protein-rich phase depends on both protein and crowder concentration, and is salt-sensitive (Fig. S5, Fig. S6). Tn2 did not phase separate with wild-type Vn or with Vn$^{DR-A50I}$, a deregulated vinculin mutant unable to bind to talin (Fig. 1f, Fig. S7a), indicating that specific talin-vinculin interactions are required for phase separation. The talin dimerization domain is also required for droplet formation, and a point mutation reducing interactions between the talin head and rod domains (Tn2$^{E1772A}$) increased the volume of the protein-rich phase, suggesting talin autoinhibition lowers its propensity to phase separate (Fig. 1g, Fig. S7b). However, neither Tn2$^{E1772A}$ alone nor Tn2 with Vn$^{D1}$, the vinculin fragment containing the talin binding site, were sufficient for phase separation (Fig. S7b).

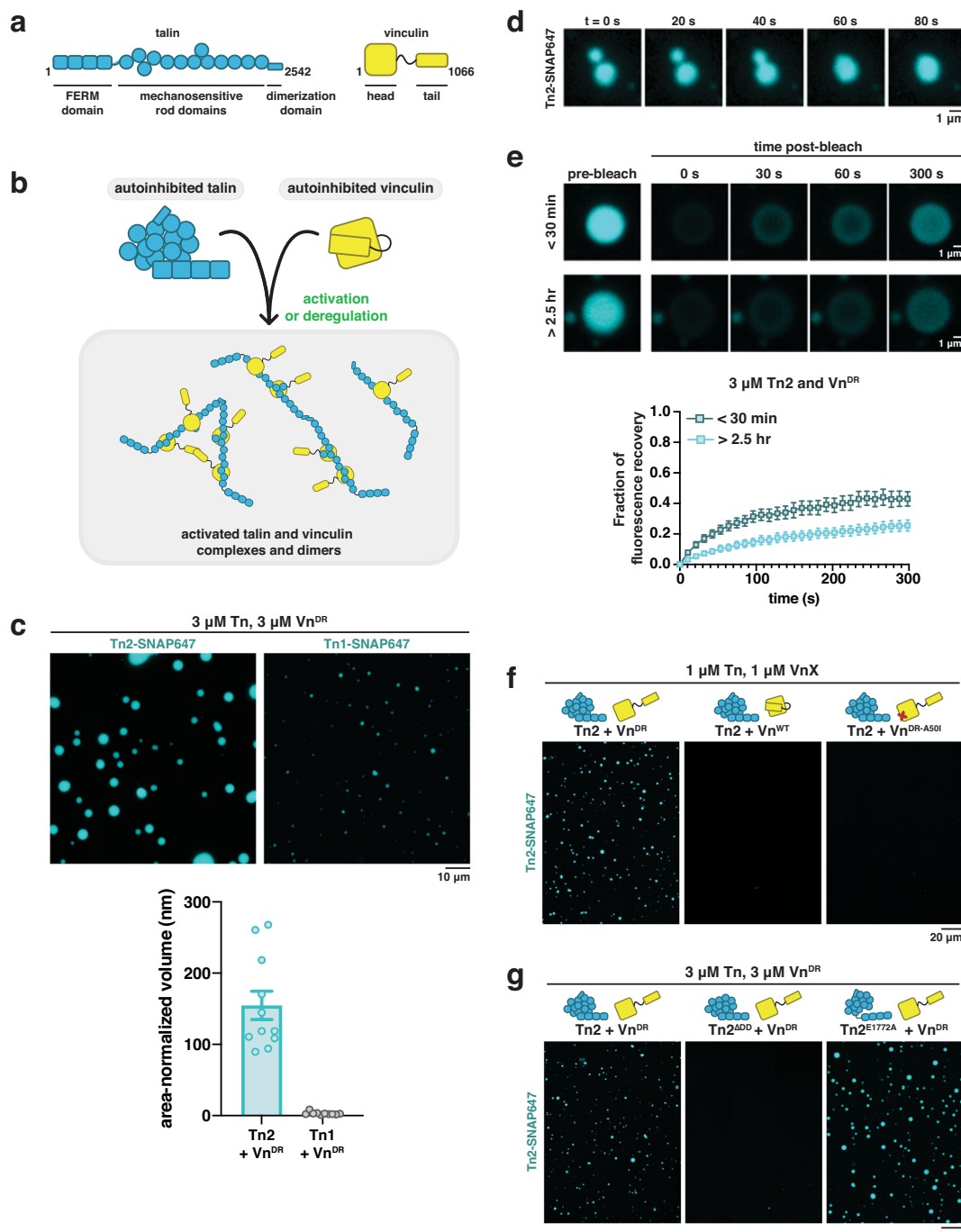

**Fig. 1 | Talin undergoes liquid-liquid phase separation. a** Domain schematics for talin and vinculin. **b** Protein autoinhibition needs to be released or removed in order for complexes to form. **c** Talin2 (Tn2) forms droplets when incubated with deregulated vinculin (Vn^DR) under crowded conditions at room temperature. Talin1 (Tn1) forms smaller droplets under the same conditions. Data are shown as mean values +/− SEM. For each condition, n ≥ 10 regions were analysed from 3 different samples. **d** Fusion of Tn2 and Vn^DR droplets demonstrate liquid-like behavior. **e** Photobleaching of Tn2-Vn^DR droplets indicates fluorescence recovery. Recovery slows as the droplets age. Data are shown as mean values +/− SEM. For each conditions, n ≥ 6 FRAP experiments were performed from 3 different samples. **f** Tn2-Vn^DR droplet formation can be disrupted by mutation of the talin binding site in vinculin (A50I). **g** Removing the talin dimerization domain (Tn2^ΔDD) reduces droplet formation, while reducing talin autoinhibitory interactions between the rod and head regions (E1772A mutation) increases phase separation. All conditions for (**c**, **f**, **g**) were repeated in triplicate, and imaged after incubation at room temperature for one hour. See Supplementary Fig. S7 for further quantification and related additional experiments. Purified proteins were mixed in the following buffer (10 mM imidazole, 50 mM KCl, 1 mM MgCl₂, 1 mM EGTA, 0.2 mM ATP, pH 7.5) supplemented with 15 mM glucose, 20 μg/mL catalase, 100 μg/mL glucose oxidase, 1 mM DTT and 0.25% methyl-cellulose (4000 cp). Error bars represent standard error.

## Talin activation drives phase separation

Integrin receptors are the foundation of focal adhesions, connecting mechanosensitive and signaling machinery within the cell to ligands in the extracellular environment. Integrins are made up of non-covalent heterodimers, referred to as the α- and β-subunits, which switch between a bent, inactive form and an extended, engaged, active conformation when bound to an extracellular ligand[36]. The β-integrin cytoplasmic domain binds directly to the talin head via two distinct

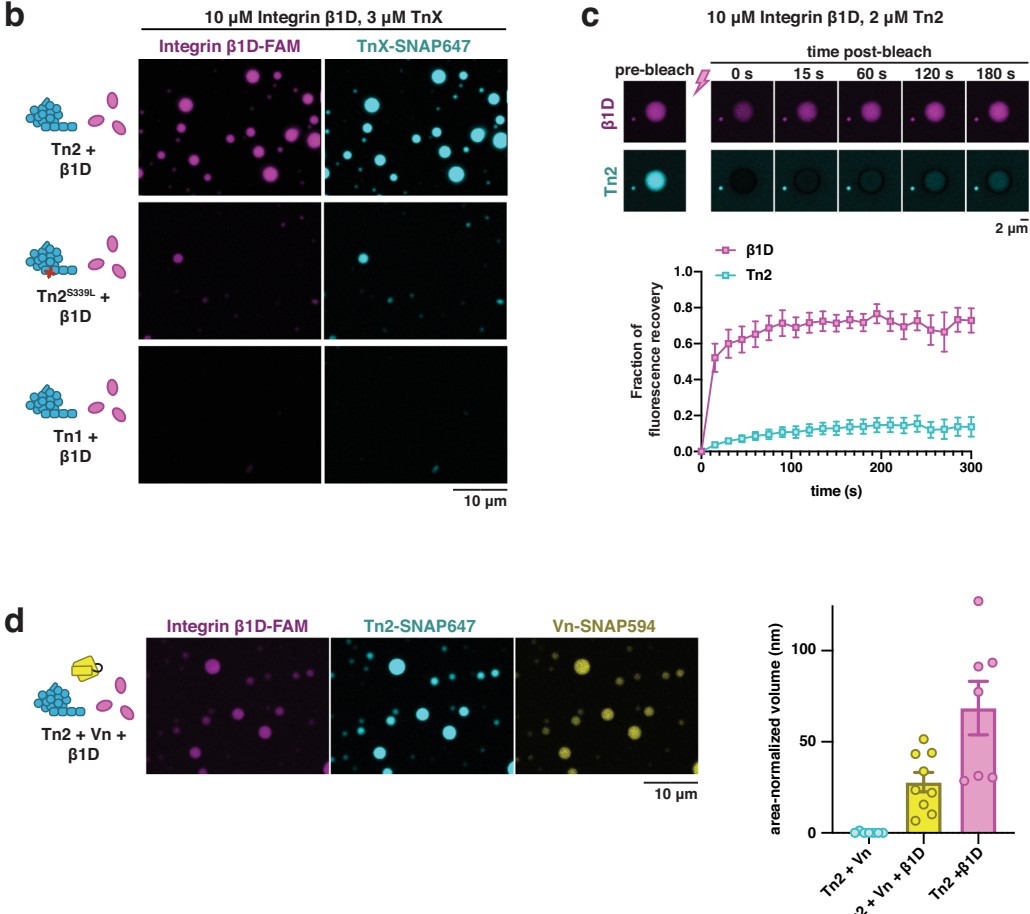

**Fig. 2 | Integrin β1D drives phase separation of talin. a** Synthetic Integrin β1D cytoplasmic tail peptide labeled with carboxyfluorescein (FAM). **b** The β1D peptide specifically drives phase separation of Tn2, not Tn1, under crowded conditions at room temperature. Phase separation can be disrupted by a mutation in the β1D binding site (TnS339L). **c** Photobleaching of β1D-Tn2 droplets indicate that Tn2 and the β1D peptide have different recovery rates. Data are shown as mean values +/− SEM. For each conditions, n = 5 FRAP experiments were performed from 3 different samples. **d** Wild-type vinculin is recruited to Tn2-β1D droplets, but slightly

decreases the amount of total phase separated material. Purified proteins were mixed in  buffer with 10 mM imidazole, 50 mM KCl, 1 mM MgCl₂, 1 mM EGTA, 0.2 mM ATP, pH 7.5, supplemented with 15 mM glucose, 20 μg/mL catalase, 100 μg/mL glucose oxidase, 1 mM DTT and 0.25% methyl-cellulose (4000 cp). All samples were incubated at room temperature for one hour before imaging. Data are shown as mean values +/− SEM. For each condition, three separate experimental runs were performed, and two or more regions within each sample were quantified.

interaction sites in the F3 domain[37,38]. This interaction is critical for integrin activation and signaling, as talin binding triggers a conformational change separating the α- and β-subunits, thereby increasing integrin's affinity for extracellular ligands[39–41]. The interaction between talin and integrin may also play a role in talin activation and engagement within focal adhesions, though experimental data for this hypothesis is lacking[42]. To test whether integrin binding can promote phase separation of talin, we synthesized a fluorescently-labeled peptide of the β1D integrin cytoplasmic tail sequence (Fig. 2a). The Tn2-integrin β1D interaction is the strongest of the characterized talin-integrin receptor interactions ($K_d$ ~ 30 μM)[43]. We observed that Tn2 formed liquid-like droplets with the β1D peptide alone under crowded conditions (Fig. 2b), without an additional activating binding partner such as VnDR, suggesting that β1D is sufficient to activate Tn2. Phase separation was dependent on the concentration of the β1D peptide, which was required in excess (Fig. S8). No phase separation was observed when Tn1 and β1D were mixed (Fig. 2b). Mutation of the

residue responsible for the increased affinity of talin2 for β1D domain (TnS339L) severely reduced the volume of the protein-rich phase, confiming that droplet formation is the result of specific talin-integrin interactions (Fig. 2b, Fig. S9)[44]. We observe that β1D is concentrated within Tn2 condensates, suggesting that talin phase separation could act to induce integrin clustering[45]. In these droplets Tn2 recovery after photobleaching was reduced compared to Tn2-VnDR droplets, while β1D rapidly recovered (Fig. 2c), suggesting that dynamics within Tn2-based condensates can vary based on binding partners. Finally, wild-type Vn is recruited to Tn2-β1D droplets, suggesting that Tn2 is in an open conformation. The total volume of the protein-rich phase was reduced in the presence of wild-type Vn, perhaps indicating a regulatory role wherein vinculin limits the extent of talin-induced integrin clustering (Fig. 2d, Fig. S10).

Together, these results suggest that activation of Tn by either VnDR or β1D is sufficient to promote talin phase separation. In addition, we found that inducing talin activation with salt also promotes talin phase

separation (Fig. S11). As previously shown, salt can artificially induce talin activation in vitro by disrupting weak interactions between rod domains, thereby triggering an "open" conformation[20]. Thus, we conclude that releasing talin autoinhibition consistently leads to its phase separation.

While our experiments suggest that both activated vinculin and β1 integrin can release talin autoinhibition, they are unlikely to be the primary talin activator during focal adhesion initiation and assembly. Instead, the phosphoinositide PIP₂ plays a major role in regulating talin localization and activation at the plasma membrane, and PIP₂ is necessary for proper formation of functional focal adhesions[46]. When autoinhibited, talin's integrin-binding sites, as well as actin and vinculin binding sites within the rod domains, are obscured. The F2F3 domains of the talin head domain have a strong preference for PIP₂, even when compared to other acidic phospholipids[47,48]. Binding to PIP₂ is predicted to trigger a shift from a globular, inactive conformation to an open, extended conformation, revealing the integrin binding sites within the F3 domain. The talin rod domains would then be released from interactions with the talin head domain, and available to recruit vinculin and actin to membrane surfaces[20,31,49]. Additionally, talin-membrane interactions are likely required to trigger the conformational change leading to integrin activation[39]. Due to the close link between PIP₂ and talin engagement within FAs, we wished to test whether PIP₂-containing membranes can also trigger phase separation by releasing talin autoinhibition. We extended our microscopy assay with phase separating protein solutions to include lipids as additional binding partners in the reaction mix (Fig. 3a). To this end, we prepared relatively homogeneous dispersions of small unilamellar vesicles (SUVs) (<100 nm in diameter) and mixed them with Tn2 under physiological conditions with a crowding agent. We observed phase separation of Tn2 when combined with a dispersion of SUVs with PIP₂-rich membranes, but not when mixed with SUVs lacking PIP₂ (Fig. 3b, Fig. S12). Phase separation of Tn2 was enhanced by including either the β1D peptide or VnᴰᴿR. Tn1 did not form droplets when mixed with 20% PIP₂ SUVs (Fig. 3c), consistent with the lower affinity of Tn1 for PIP₂[50]. Condensates with Tn2, SUVs and β1D were much more viscous than droplets in our earlier experiments and appear as assemblies of spherical droplets that clump together (Fig. S13), indicative of rapid aging. Even more strikingly, these condensates did not require macromolecular crowding conditions to form and appeared to have a stronger tendency to phase separate than with either talin activator independently.

### Talin phase separates on supported lipid bilayers

Next, we added Tn2 to fluid supported lipid bilayers (SLBs) containing 5% PIP₂ (Fig. S14), to investigate phase separation of talin on a membrane geometry that more closely resembles that of living cells. Tn2 alone was not sufficient to form clusters on PIP₂ containing bilayers (Fig. S15). However, when added together with wild-type Vn, micron and sub-micron sized two-dimensional (2D) circular Tn clusters form rapidly (<5 min) (Fig. 3d). Fluorescence signals of labeled talin, vinculin and PIP₂ colocalize and indicate collective dilute and dense phases on/in the membrane (Fig. S16, Fig. 3d). The clusters form in the absence of macromolecular crowding, but require a minimum amount of PIP₂ in the membrane to form (Fig. S17), and recover partially after photobleaching (Fig. 3e, Fig. S18). In addition, clustering requires the talin dimerization domain, suggesting a mechanism analogous to 3D droplet formation in solution under crowded conditions (Fig. S15). The clusters grow over time, with fluctuating boundaries (Fig. 3f, Fig. S16c-e). These observations suggest that while the clusters may be less fluid than their 3D counterparts, they are not static structures. In the presence of polymerizing actin, the clusters take on more irregular shapes, but do not seem to recruit actin filaments or mediate polymerization (Fig. S20). Occasionally, the proteins

undergo spinodal decomposition, a phase separation mechanism that does not require nucleation, resulting in characteristic patterns instead of circular condensates (Fig. S21). The β1D peptide is strongly recruited to the clusters, at a much lower talin-to-peptide ratio than required in solution under crowded conditions (Fig. 3g). While clustering of transmembrane, alpha-beta subunits of integrin receptors is undoubtedly a more complex process, the ability of membrane-bound talin to recruit the β1D peptide to the membrane and form 2D, phase separated micro-domains indicates that it could have a similar effect on full-length integrin receptors. Integrin-talin clusters are thought to be precursors of FAs, and are dependent on cellular PIP₂ levels, supporting the idea that PIP₂-talin interactions play a role in integrin clustering and FA assembly[45].

Figure 4 illustrates a plausible mechanism of formation for 2D biomolecular condensates made from talin and vinculin, based on our results. Talin binds to and is activated by PIP₂ in the membrane, triggering talin dimerization and recruitment of wild-type vinculin. Once bound to talin, vinculin also forms a dimer, thereby recruiting additional membrane-bound talin molecules and leading to a network of multivalent interactions, resulting in the formation of membrane-bound talin-vinculin condensates, and corresponding PIP₂-enriched membrane domains as their bases within the bilayer.

### Clustering leads to high talin-membrane interaction strength

To elucidate the interactions between talin and the membrane, we developed an assay to measure the strength of single Tn2-PIP2 interactions using optical tweezers (Fig. 5a). Previous force spectroscopy experiments have shown that both talin-vinculin and talin-integrin interactions can be mediated or strengthened through force application to the talin rod domain[51–53]. However, none of these experiments have explored the effect of force in the context of the full-length, autoinhibited talin protein. Talin's N-terminal FERM domain secures the talin rod in a globular, inactive conformation in solution[20]. We propose that upon binding to PIP2 in the plasma membrane, the FERM domain releases the talin rod domain, making it available for force transduction.

To test whether the talin rod has the same mechanosensitive response to tension when bound to the membrane through the FERM domain, we used a dual trap optical tweezer set-up to pull on membrane-bound full-length talin. This assay required two micron-sized beads: one bead coated with PIP2-containing lipids and a second with Tn2 bound via a DNA handle and biotin-neutravidin interactions (Fig. 5a).

Our force spectroscopy results suggest that multiple Tn2 molecules usually form tethers between the beads. Force curves show multiple, sequential rupture events (Fig. S24b). Often interactions were strong enough to pull the membrane-coated bead out of its optical trap. We hypothesized that the strength of talin-membrane interactions may be due to talin's propensity to phase separate when bound to the membrane. To test this, we used Tn2ᐃᴰᴰ, thereby retaining the membrane affinity and mechanosensitivity of full-length Tn2, while eliminating potential clustering on the membrane. Indeed, measurements for Tn2ᐃᴰᴰ samples resembled models of force curves of single proteins, and thus allowed more thorough quantification. Rupture forces for Tn2ᐃᴰᴰ-membrane interactions ranged from 1 to 60 pN, with a mean rupture force of 28.72 pN (n = 104 rupture events) (Fig. 5b). These results demonstrate that Tn2-membrane interactions can withstand forces within the range at which talin rod domains unfold[54]. We also observed consecutive unfolding/refolding cycles, with two distinct, consistent, and reversible unfolding events (Fig. 5c). These unfolding events occur at forces consistent with reversible unfolding events measured for the talin rod domain alone, i.e., between 5 and 25 pN[51,52,54]. This indicates that talin-membrane interactions are strong enough to withstand the forces required to

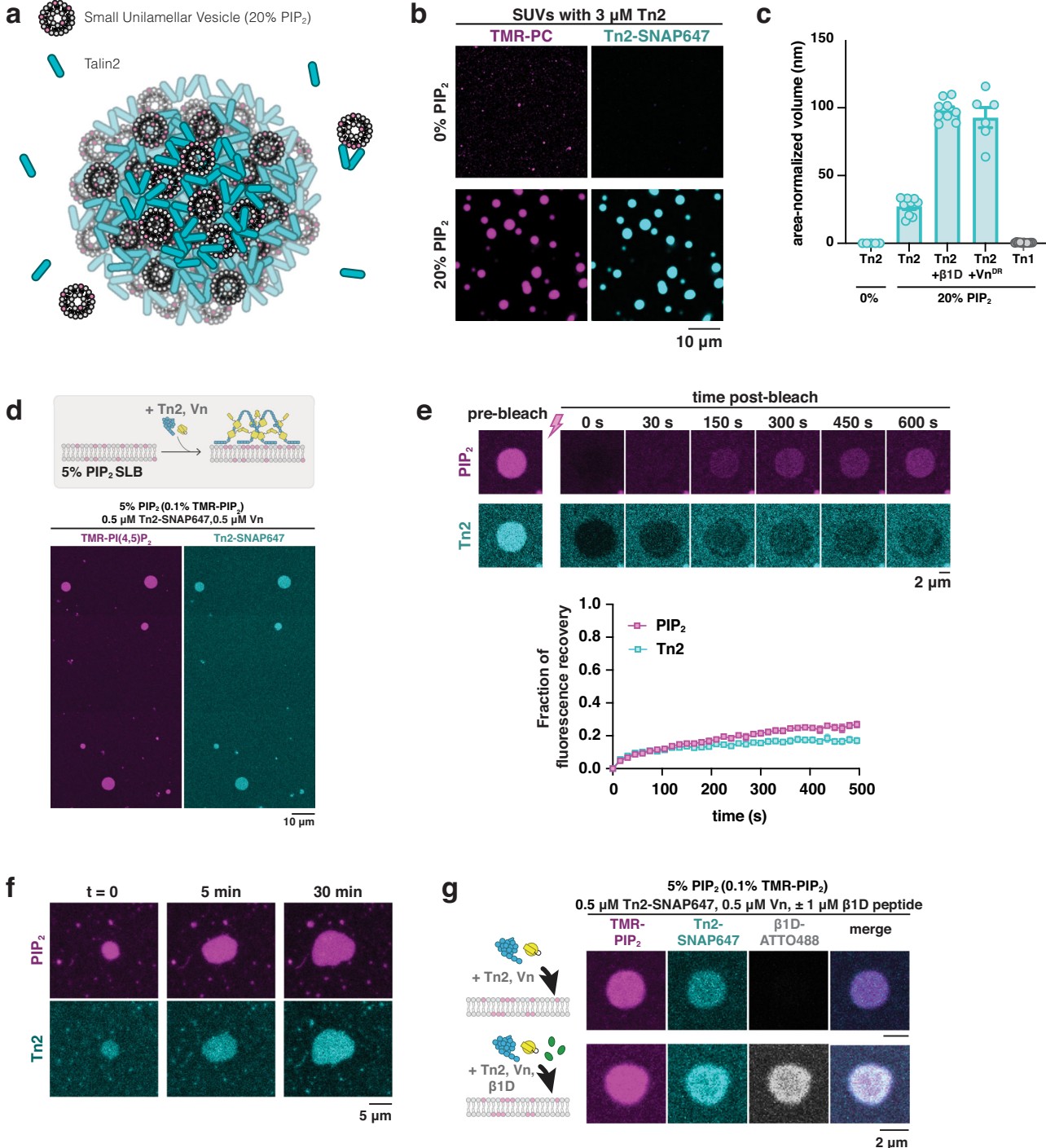

**Fig. 3 | Membrane-bound talin phase separates. a** Schematic showing phase separation of Tn2 with PIP$_2$-containing SUVs. **b** Tn2 forms droplets in the presence of PIP$_2$-rich vesicles under crowded conditions, but not in the absence of PIP$_2$. Top: SUVs consisting of 75% POPC, 15% POPE, 10% POPS, Bottom: SUVs consisting of 55% POPC, 20% PIP$_2$, 15% POPE, 10% POPS. Note that vesicle sizes are below the resolution limit; visible structures are liquid condensates. **c** Phase separation in the presence of vesicles is enhanced by the addition of other phase separation binding partners of talin, β1D and Vn$^{DR}$. Data are shown as mean values +/- SEM. For each condition, between n = 5 and n = 13 regions were analyzed from 3 different experimental samples. **d** Talin forms 2D clusters on PIP$_2$-rich supported lipid bilayers (SLBs). **e** Tn2-PIP$_2$ clusters recover fluorescence slowly after photobleaching. Data are shown as mean values +/− SEM. n = 10 FRAP experiments were quantified from 3 different experimental samples. **f** Tn2-PIP$_2$ clusters change shape and grow over time. **g** Integrin β1D peptide is also recruited to Tn2-PIP$_2$ clusters on the membrane. All lipid-based experiments were carried out in buffer with 10 mM HEPES pH 7.5, 100 mM NaCl, either with 0.25% methyl cellulose (vesicle experiments) or without crowding reagent (SLB experiments).

unfold the individual talin rod domains. Thus, the talin rod domains display similar mechanosensitive behavior in the context of the full-length, membrane-bound protein, when compared to the isolated rod domains[54].

## Discussion

Strong evidence points to talin acting as a master regulator of focal adhesion assembly and organization. We have shown in this study that activation of talin through multiple pathways triggers the formation of

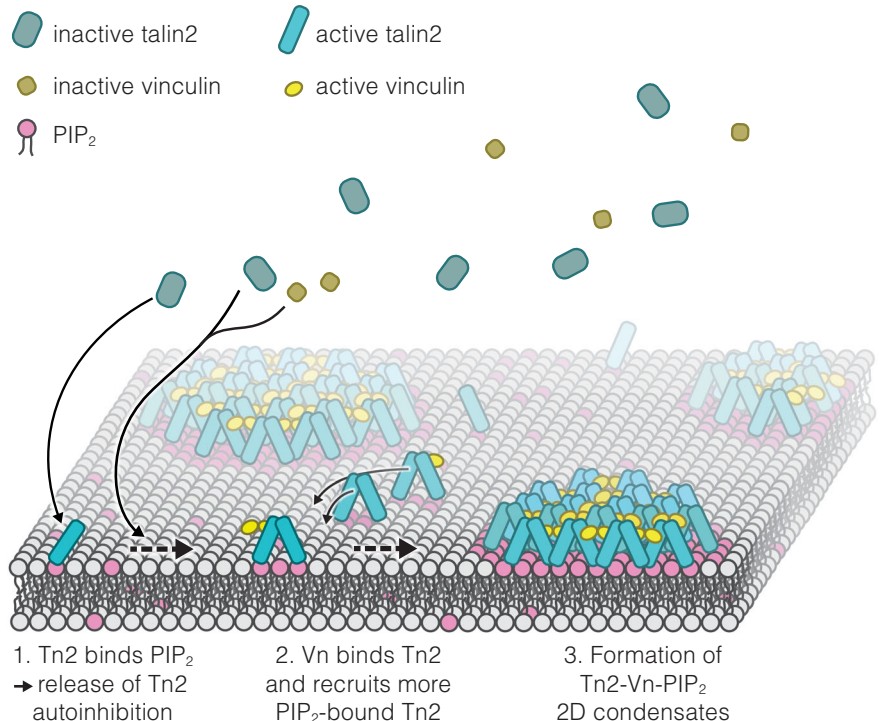

**Fig. 4 | Proposed mechanism of Talin phase separation on PIP₂-containing SLBs.** Talin is recruited to the membrane by the phospholipid PIP₂, which also activates the protein. This allows for the dimerization of talin and the binding of vinculin. Vinculin can also form dimers, resulting in multivalent interactions and the recruitment of more PIP₂-bound (i.e., activated) talin, eventually leading to the formation of Tn2-Vn 2D condensates on the membrane and PIP₂-enriched domains in the membrane. In solution these proteins are not able to interact. Binding to the membrane triggers a cascade of protein activation and interactions that lead to liquid-liquid phase separation into 2D clusters.

LLPS condensates in vitro. Focal adhesions show liquid-like properties in cells, and share many characteristics with confirmed biomolecular condensates. However, focal adhesions are naturally highly anisotropic: they exhibit these liquid-like properties in two dimensions laterally along the membrane, but show a highly structured organization orthogonal to the membrane[16] in large part due to talin's role as a core structural scaffold[19]. Our in vitro experiments suggest that these properties could be explained by lipid-regulated phase separation, thus restricting the formation of typically 3-dimensional condensates to a two dimensional membrane.

Other recent work has shown the wetting of protein condensates on membranes[21–23], and a lowering of the phase separation threshold by membrane binding[24–27] or clustering of transmembrane proteins[11,12,28,55,56]. However, we propose a regulatory mechanism in which lipids constitute an interaction partner required for the condensation of proteins, thereby inducing phase separation of proteins from solution into truly 2-dimensional condensates with a preferred protein orientation. Hence, this promotes a structured organization of focal adhesions in the remaining third dimension, i.e., into the cytosol, as a spatial cue for cytosolic processes. Thus, talin's role as a scaffold and the extended length of its rod domains allow for a high level of organization within focal adhesions, while retaining the desirable traits of dynamic, liquid-like condensates.

Biomolecular condensates have been identified in a range of physical states; varying greatly in terms of dynamics, stability, lifespan, and function. The talin condensates we observe in solution become more viscous over time, and protein turnover varies depending on components and composition, which is in agreement with other LLPS experimental and theoretical literature[35,57–60]. Even the 3D droplets that form in combination with PIP₂ SUVs, the most viscous droplets we observe, have a characteristic rounded shape indicative of LLPS, which suggests that droplets were originally much less viscous, but aged over time. Similarly, the recovery of 2D membrane-associated

Tn2 condensates on PIP₂ SLBs is fairly slow with a high immobile fraction, similar to what has been described as bioreactive gels[58]. This would be consistent with talin's low diffusivity laterally along the membrane in focal adhesions[61]. The mechanosensitive nature of focal adhesions may require this increased viscosity, in order to withstand forces applied through actomyosin contraction. A similar explanation has been offered for the gel-like state of the centrosomal condensates, which must withstand forces applied by the spindle during mitosis[58,62].

The mechanism we suggest (Fig. 4), involves the activation of talin by PIP₂-binding, which then allows for the dimerization of talin and binding of wild-type vinculin reminiscent of allosteric regulation. We suggest that the ability of both proteins to form dimers and bind each other results in a network of interactions, triggering the formation of liquid-liquid condensates. Since the talin-activator, the lipid PIP₂, is a component of a 2D membrane, the formation of the resulting condensate is strictly limited in space to the membrane surface.

Our other experiments reinforce the hypothesis that release of talin autoinhibition is critical for triggering phase separation. In Fig. 1, we show that our deregulated vinculin mutant (VnDR) can act both as a talin-activator, and lead to the formation of condensates through multivalent interactions. In Fig. 2, we show the same for the integrin peptide β1D. Our hypothesis that this reconstitutes talin activation is backed by the observation that in the presence of β1D, talin is able to recruit wild-type vinculin (VnWT) to condensates. In Fig. 3a–c, we observe that PIP₂ also releases talin inhibition. In the form of SUVs, this further allows for multivalent binding and hence, the formation of condensates. However, in the case of a continuous membrane, such as SLBs or the plasma membrane, the latter effect can not take place, because PIP₂ lipids are free to diffuse laterally and therefore do not mediate interactions between talin dimers. Here, PIP₂ only acts as a talin-activator and the mechanism relies on vinculin for intermolecular attraction.

We further demonstrate that individual talin-membrane interactions are strong enough to withstand forces capable of inducing rod

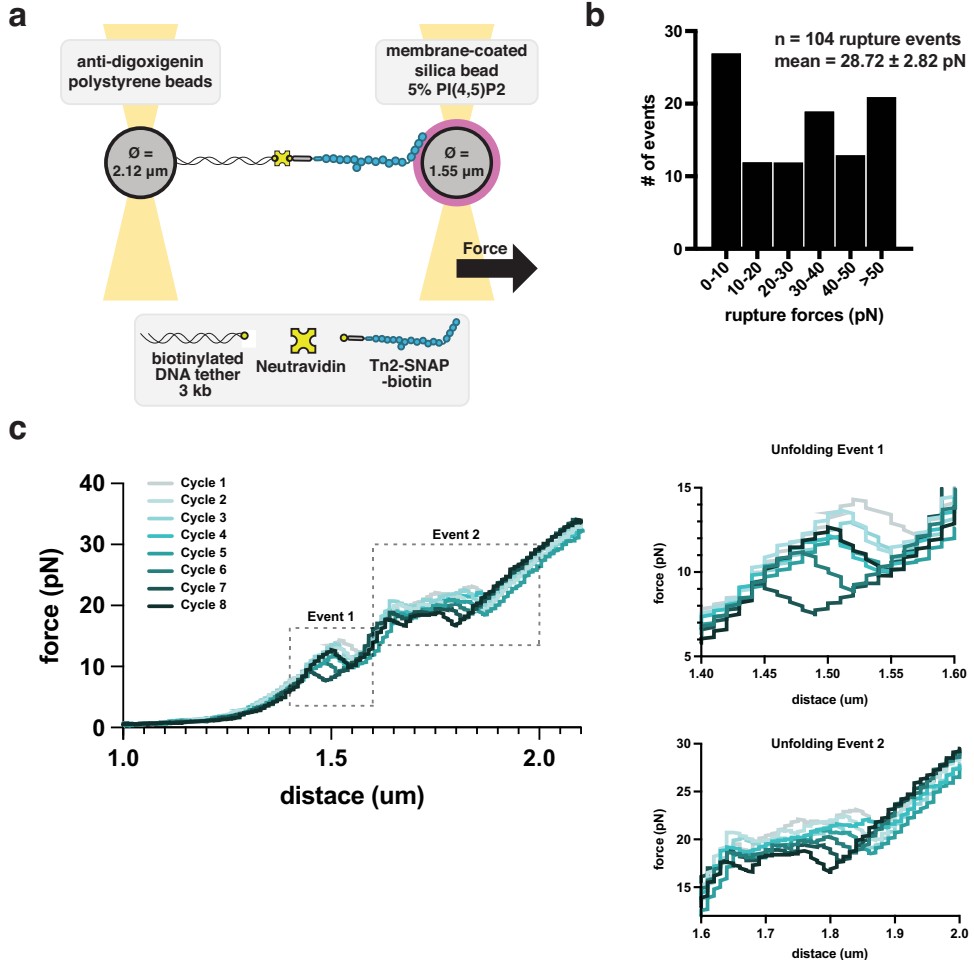

**Fig. 5 | Clustering strengthens talin-membrane connections. a** Optical trap set up for measuring single talin-membrane interactions. **b** Rupture force distribution for interaction of Tn2$^{\Delta DD}$ with PIP$_2$ bilayer. We measured the forces at which the protein detaches from the membrane. **c** Tn2$^{\Delta DD}$ force curve pre-rupture showing consecutive protein unfolding events. We can identify two major unfolding domain unfolding events. Our measurements of the wild-type Tn2 also suggest that the ability to dimerize strengthens talin-membrane interactions beyond the sensitivity of the current experimental system, which is specifically designed to measure binary interactions. An expansion of force-determining methodology will be required to reach a deeper understanding of the mutual roles of talin-talin and talin-membrane interactions in focal adhesion mechanosensation. Our results suggest that PIP$_2$-induced cooperative talin-membrane interactions are necessary for talin to resist the high mechanical forces within focal adhesions (up to 10 pN per molecule[63,64]), since the talin-integrin linkage is a low affinity interaction (30–500 μM)[43]. Additionally, kindlin may further cooperate with talin and PIP$_2$ to enable transmission of high mechanical force at focal adhesions[65].

As more biomolecular condensates are identified in diverse cellular processes, it is becoming clear that interactions with membranes play a much more general role in their formation, function, and regulation[66–71]. Until now, lipid bilayers have been viewed mostly as passive surfaces which can spatially target and increase phase separation through physical effects. Here, we argue that specific interactions with phospholipids can regulate protein phase separation: PIP$_2$ not only concentrates talin at the membrane, but also releases talin autoinhibition. By activating talin, PIP$_2$ likely triggers phase separation and enables the recruitment of other adhesion proteins, thereby initiating a crucial step in focal adhesion assembly. PIP$_2$ binds

events, which likely each represent multiple individual rod domains unfolding within full-length talin. The force range at which these unfolding events occur, as well as the reversibility of the events when released from tension, is in agreement with force curves for the rod domain alone[47]. All optical-trap experiments were carried out in the following buffer: 10 mM HEPES pH 7.5, 100 mM NaCl.

and regulates additional focal adhesion adapter proteins, including FAK and vinculin, suggesting that lipids could coordinate the recruitment and phase separation of additional components. Our experiments provide evidence that spatial organization on the level of lipids could have structural effects that reach far into the cytoplasm.

## Methods
### Proteins
Plasmids for Talin2, Vn, Vn$^{3Q}$ and Vn$^{DR}$ were previously made for ref. 31. We refer to these here as pET21_hTalin2 L435G −3C-SNAP-His, pET21_hVinculin-3C-SNAP-His, pET21_hVinculin K944Q R945Q K1061Q-SNAP-His and pET21_hVinculin K944Q R945Q K1061Q-SNAP-His. All Talin2 plasmids contain the L435G mutation to reduce calpain cleavage.

To generate plasmids for expressing Talin1 as his-SNAP-tagged fusions in E. coli, ORFs were PCR amplified from pET101-hTalin1-His[20] and assembled with the PCR amplified backbone from pET21_hTalin2 L435G −3C-His[31] using seamless cloning (ThermoFisher Scientific/Invitrogen™ GeneArt™ Seamless Cloning and Assembly Enzyme Mix), resulting in pET21_hTalin1-3C-SNAP-His.

Site-directed mutagenesis was performed on pET21_hTalin2 L435G-3C-SNAP-His[31] to generate the S339L and E1772A mutants and assembled with homologous recombination or Seamless Cloning

(GeneArt™) respectively, resulting in pET21_hTalin2 L435G E1772A-3C-SNAP-His and pET21_hTalin2 L435G S339L-3C-SNAP-His.

The plasmid for the ΔDD Tn2 truncation was amplified from pET21_hTalin2 L435G-3C-SNAP-His[31] resulting in pET21_hTalin2 L435G 1-2493 −3C-SNAP-His.

For Vn^DR, site directed mutagenesis was performed on pET21_h-Vinculin K944Q R945Q K1061Q-SNAP-His[31] to introduce the additional N773A and E775A mutations and then assembled using Gibson Assembly[72], resulting in pET21_hVinculin N773A E775A K944Q R945Q K1061Q-SNAP-His. Site directed mutagenesis then also was performed on the resulting plasmid to introduce the A50I mutation and assembled with homologous recombination, resulting in pET21_hVinculin A50I N773A E775A K944Q R945Q K1061Q-SNAP-His.

The plasmid for the Vn^D1 fragment was amplified from pET21_hVinculin-3C-SNAP-His[31] and assembled by blunt-end cloning, resulting in pET21_hVinculin 1-258 −3C-SNAP-His.

### Protein expression and purification
Constructs were expressed in *E. coli* BL21 (DE3) gold using ZY auto-induction medium. Talin proteins were all purified using the same protocol, based on that described in a previous report[20]. Cells were lysed by sonication in 50 mM Tris-HCl pH 7.8, 500 mM NaCl, 5 mM imidazole, 3 mM β-mercaptoethanol, 1 mM EDTA, and Roche cOmplete protease inhibitor tablets (Roche, Basel, Switzerland), followed by purification using nickel-affinity chromatography (cOmplete His-Tag purification column, Roche), and cation exchange (HiTrap SP FF, GE Healthcare, Chicago, Illinois). Next, the his-tag was either removed using overnight incubation with 3 C protease, or labeled using overnight incubation with SNAP-AlexaFluor647 (New England Biolabs, Ipswich, Massachusetts). Finally, protein was further purified by size-exclusion chromatography using either a Superdex 200 16/600 column (GE Healthcare) or Superose 6 10/300 column (GE Healthcare) in 50 mM HEPES pH 7.8, 150 mM KCl, 3 mM β-mercaptoethanol, 1 mM EDTA, and 10% glycerol, followed by flash freezing for storage at −80 °C.

For vinculin proteins, cells were lysed by sonication in 50 mM Tris-HCl pH 7.8, 500 mM NaCl, 5 mM imidazole, 3 mM β-mercaptoethanol, 1 mM EDTA, and Roche cOmplete protease inhibitor tablets. Following lysis, TritonX-100 was added for a final amount of 1% by volume. Full-length vinculin cell lysates were incubated on Roche cOmplete His-Tag resin for 2 hr at 4 °C, then washed with 50 mM Tris-HCl pH 7.8, 500 mM NaCl, 10 mM imidazole, 3 mM β-mercaptoethanol, 1 mM EDTA. After washing, proteins were incubated overnight with either 3 C protease to remove the SNAP-his tag, or labeled with SNAP-AlexaFluor488 or SNAP-Surface594 (New England Biolabs). Following removal or elution from beads, vinculin proteins were then further purified by size-exclusion chromatography using Superdex 200 16/600 column (GE Healthcare) or Superose 6 10/300 column (GE Healthcare) in 50 mM HEPES pH 7.8, 150 mM KCl, 3 mM β-mercaptoethanol, 1 mM EDTA. Vinculin fragments were purified using nickel-affinity chromatography, immediately eluted from the column with 1 M imidazole, cleaved overnight with 3 C, and separated from the cleaved SNAP-his tag by reverse nickel-affinity chromatography. This was followed by size-exclusion chromatography using a Superdex 75 10/300 in 50 mM HEPES pH 7.8, 150 mM KCl, 3 mM β-mercaptoethanol, 1 mM EDTA. Proteins were flash frozen and stored at −80 °C. We found that vinculin which is fluorescently labeled via SNAP tag affects membrane condensate formation and dynamics, hence we avoided fluorescently labeled vinculin on most experiments.

Actin proteins were purchased in lyophilized form from HYPERMOL.

### β1D Peptide
The fluorescently labeled β1D peptide was produced by the Bioorganic Chemistry & Biophysics Core Facility at the Max-Planck-Institute of Biochemistry by solid phase synthesis using Fmoc/*t*Bu chemistry and microwave heating on a Liberty Blue peptide synthesizer (CEM Corporation, Mathews, NC, U.S.A.).

### Analytical size-exclusion chromatography assays
Proteins used were first buffer exchanged or diluted to match the conditions tested (i.e., 75 mM or 500 mM KCl). Proteins were incubated together on ice for 15 min, and prespun in a tabletop micro-centrifuge at $15,000 \times g$. 75% of the sample volume was removed, avoiding any potential pellet, and applied to a Superose 6 Increase 3.2/300 column with 20 mM HEPES pH 7.8, 1 mM EDTA, 3 mM β-mercaptoethanol, and either 75 mM or 500 mM KCl.

### 3D biomolecular condensate assays
Most experiments with condensate droplets were performed in flow chambers. For this, glass coverslips were washed extensively with milliQ water and dried with nitrogen gas, then coated using a solution of 80% ethanol pH 2, 2 mg/mL methoxy-poly (ethylene glycol)-silane and dried for several hours at 75 °C. Immediately before using, coverslips were washed extensively with milliQ water, dried under a stream of nitrogen, and attached to adherent flow chambers (Ibidi, Martinsried, Germany).

For the experiments in Fig. S5 we used microtiter plates (Greiner Bio-One, 384-well glass bottom SensoPlate™), which we passivated beforehand with 50 μl of 5 mg/ml β-casein (Sigma Aldrich) for 20 min.

Protein for each experiment were first mixed together while still in stock buffer (see above) at stock concentration. All samples with condensate droplets were made in the following buffer, unless stated otherwise: 10 mM imidazole, 50 mM KCl, 1 mM MgCl2, 1 mM EGTA, 0.2 mM ATP, 0.25% methyl-cellulose, pH 7.5 supplemented with GODCAT (15 mM glucose, 20 μg/mL catalase, 100 μg/mL glucose oxidase) and 1 mM DTT. We use methyl cellulose as a crowding agent at much lower concentrations as commonly used crowders like PEG or ficoll[73]. Methyl cellulose (Thermo Scientific, 4000 cP @ 2% solution) was dissolved for a stock concentration of 1%.

After the sample was prepared and pipetted onto/into the imaging container, samples were typically incubated for one hour before imaging.

Imaging was performed with a Zeiss LSM 780/CC3 confocal microscope equipped with a C-Apochromat, 63x/1.4 W objective. PMT detectors (integration mode) were used to detect fluorescence emission (excitation at 488 nm for ATTO488, 594 nm for SNAP594 and 633 nm for SNAP647) and record confocal images. All experiments were conducted at room temperature.

### Quantification
The amount of phase separated material in confocal microscope data was quantified with a custom written code. These results are shown in Figs. 1c, 2d and 3c and Supplementary Figs. S5, S7, S8, S9 and S13. Our Matlab code first binarizes all z-slices of a confocal z-stack and determines the area of biomolecular condensates visible in each z-slice. The volume is then extrapolated by just adding up the areas of all z-slices, which are spaced 1 μm apart. We then divide the resulting volume by the total area of the field of view, to get an area-normalized volume, i.e., the average height of a droplet, equivalent to how most types of precipitation is quantified.

### Supported lipid bilayer assays
The SLBs were prepared in the same fashion as we described previously[31] and is based on a protocol by Braunger et al. [74]. Briefly, lipids were mixed in a glass vial, and dried under a continuous stream of N$_2$, then dried overnight in a vacuum chamber at room temperature. The lipid film is then gently rehydrated in citric acid buffer pH 4.8, and incubated at room temperature for 20 min before vortexing briefly.

To produce small unilamellar vesicles (SUVs), the solution is then sonicated for 30 min (30 s on/30 s off intervals).

Coverslips were cleaned using piranha solution for at least 15 min. Immediately before using, coverslips were rinsed thoroughly in deionized water and dried with $N_2$, then attached to Ibidi adherent flow chambers. To form SLBs, 60 μL of 0.2 mg/mL SUVs were added to individual flow chambers and incubated for 3 min, followed by $2 \times 80$ μl imaging buffer buffer (1x) to remove excess vesicles.

SLBs were then tested for bilayer integrity and fluidity using fluorescence recovery after photobleaching (FRAP) (Figs. S14, S19). To the fluid bilayers, proteins were added in lipid buffer (10 mM HEPES pH 7.5, 100 mM NaCl) supplemented with GODCAT (15 mM glucose, 20 μg/mL catalase, 100 μg/mL glucose oxidase).

For experiments on SLBs that include actin, we preincubate the SLBs with 0.5 mM Tn2 for 15 min. To the fluid bilayers, proteins and actin (5% ATTO488-actin) were added in imaging buffer supplemented with GODCAT.

Imaging was performed with an LSM 780/CC3 confocal microscope (Carl Zeiss, Germany) equipped with a C-Apochromat, 40×/1.2 W objective. PMT detectors (integration mode) were used to detect fluorescence emission (excitation at 488 nm for ATTO488, 594 nm for SNAP594 and 633 nm for SNAP647) and record confocal images.

## Lipid co-sedimentation assays

Lipid co-sedimentation assay were conducted as previously described[75,76]. Briefly, liposomes were swelled from dried lipid in 20 mM HEPES, pH 7.5 and 100 mM NaCl. FA proteins were mixed with 1 mg/mL liposomes and incubated at room temperature for 30 min, then spun at $18,000 \times g$ in a tabletop microcentrifuge at 4 °C. Equal volumes of pellet and supernatant were analyzed by gradient SDS-PAGE, and quantified using Fiji. For quantification, the percent of protein in the pellet of protein-alone control samples was subtracted from all experimental samples.

## Optical tweezers assay to study talin-membrane interactions.

To extract forces acting between talin and the $PIP_2$-doped lipid bilayer, we designed an assay using two optically trapped beads, one of which was functionalized with the $PIP_2$-doped lipid bilayer and talin, and the other was functionalized with short DNA handles binding to biotinylated talin structures via neutravidin (see Fig. 5a). A similar assay to probe protein-membrane interactions was previously used by Ma et al. [77].

## Preparation of functionalized beads.

Membrane-coated beads were prepared by spreading small unilamellar vesicles of defined lipid composition on silica beads as described earlier[78]. Briefly, dried lipid films are prepared and equilibrated at RT. 500 μl of buffer are added to 0.2 mg lipids, and incubated for 30 min at RT to let the films swell, vortexed until all lipid material was in suspension and sonicated for $8 \times 1$ min in ice with 1 minutes breaks until the suspension was opaque. We then centrifuged the suspension at 4 °C and 15,000 g for 5 min, to separate multilamellar material from unilamellar bilayers. We used the supernatant/unilamellar fraction to coat the silica beads. 70 μL of silica beads (Spherotech SIP-30-10 Sphero Silica Particles, 3 μm, 5% w/v) were cleaned by 3 centrifuging/washing cycles with 50 μL of buffer each (final volume approx. 70 μL). Cleaned beads (18 μl) were incubated with 220 μL vesicle suspension, vortexed for 45 min to shear additional vesicle material off the membrane-coated beads and spinned them down at 500 g for 1 h. Supernatant was removed and the pellet was resuspended with 500 μL measurement buffer and washed 2 times before using the membrane-coated bead suspension for the measurement. Coated beads were functionalized with talin by incubating 6 μL of bead solution with 2 μL of 0.5 μM Talin solution for 1 min at RT. After incubation, the bead suspension was diluted to 1 mL with measurement buffer and used for the experiment.

Beads with DNA handles were prepared by incubating 1 μL of anti-digoxigenin functionalized polystyrene beads (Spherotech Anti-dig-coated polystyrene beads, 2.12 um, 0.1 w/v%) with 1 μL 3 kb DNA handles (3xdig/3 kb DNA/3xbiotin handle, approx. 20 ng/mL).The functionalized beads were then incubated with 3 μl of 1 mg/mL neutravidin solution for 1 min. After incubation, the bead suspension was diluted to 1 mL with measurement buffer and used for the experiment.

## Setting up the optical tweezers assay.

To setup the assay, we used a LUMICKS flow cell and microfluidic system in a confocal C-trap setup (LUMICKS) that allows for recording force and fluorescence data in parallel. To probe protein-membrane interaction forces, we used the following workflow: first, we trapped a membrane-coated bead in one trap and a DNA handle-functionalized bead in the second trap. We used the confocal scanning modality of the instrument to confirm that the fluorescently labeled lipid bilayer is homogeneously spread on the silica beads. We established the binding between DNA handle to talin on the membrane-coated beads by approaching the two beads. Subsequently, we recorded force-extension curves (FEC) by moving the bead in the second trap away from the bead in the first trap and read out interaction forces acting between talin and membrane. Since rupture forces typically depend on the speed at which the optical trap moves, we ensured a constant speed of 0.25 um/s throughout the experiments.

Before studying the full protein construct, we ran a sequence of control measurements (Fig. S22) and considered models (Fig. S23). We tested (i) the 3 kb DNA handle by directly binding it to one anti-digoxigenin and one streptavidin-coated bead and recording typical DNA melting curves. We optimized the DNA to having a single DNA handle between the beads. (ii) The interaction of the DNA handle with a 0% $PIP_2$ lipid bilayer, as well as the (iii) the interaction of the DNA handle streptavidin construct with the 0% and 5% $PIP_2$-doped membrane without the protein.

## Analysis of optical tweezers data.

Before analyzing FECs, we first subtracted the background force: we recorded an FEC without tether and subtracted that from the final data. High resolution distance data (all force and trap data were recorded at 78 kHz) were obtained by looking at the trap-trap distance and subtracting the bead displacement. For plotting purposes, we downsampled the FEC by a factor of 500. An example code showing how we obtained the high-resolution data and subtracted the background can be found here: https://harbor.lumicks.com/single-script/b7b98127-1b09-4505-9967-ff1c0b2aaf92.

We were interested in the rupture force of a single DNA tether and one (or more) talin complexes bound to the membrane. Therefore, the protocol for extracting rupture forces from FECs was as follows: (1) If the tether had multiple force jumps, we would only extract the value of the last jump, when the force jumps to zero, indicating that the protein handle is fully detached from the membrane. (2) We monitored not only the force at which a rupture occurs, but also the distance. The distance allowed us to compare various constructs. (3) We ignored rupture force values above 60 pN. A single strand of dsDNA melts at 60 pN, so rupture forces above this value would indicate the presence of multiple DNA handles bound to the functionalized membrane.

## Statistics and reproducibility

Data in bar graphs is always shown as mean values +/− SEM and individual datapoints are superimposed. Data is always taken from at least 3 different samples. FRAP experiments in main text also show mean values +/− SEM from at least 3 different samples. Some Supplementary Figs. show single FRAP experiments and are marked as such.

Observations in microscopy images were always repeated in triplicate with similar results. For our main figures, in most cases this additional data is included in accompanying types of quantification in the form of bar graphs or FRAP curves, either within the same figure, or

in a Supplementary Fig. which is mentioned in the figure legend. Only exceptions to this are Fig. 3d, f, g, which were, however, also performed in triplicate with similar observations.

As a supplementary Fig. S22 we include our observations of spinodal decomposition of Tn2 on $PIP_2$ membranes, which we found difficult to reproduce, with only two experiments in which we saw similar phenotypes. We also mention this in the figure legend.

## Reporting summary

Further information on research design is available in the Nature Portfolio Reporting Summary linked to this article.

## Data availability

The data that support the findings of this study are available from the corresponding author upon request. Source data are provided with this paper.

## Code availability

A simple Matlab script is used to analyze the amount of phase separated material and is available from the corresponding author upon request.

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

## Acknowledgements

We thank Michaela Schaper for help by cloning plasmids and we thank Stefan Übel for synthesizing the β1D integrin peptide. Experimental support, microscopy equipment, and analysis of the optical trap experiments were provided by LUMICKS (Amsterdam, Netherlands). A special thanks to Bärbel Lorenz for helping to conceive and realize the optical trap experiments and Aafke van den Berg for data analysis. This work is part of the MaxSynBio consortium, which is jointly funded by the Federal Ministry of Education and Research of Germany and the Max Planck Society. C.F.K. is a recipient of the Humboldt Research Fellowship for Postdoctoral Researchers and has received funding from the European Union's Horizon 2020 research and innovation program under the Marie Sklodowska-Curie grant agreement No 794162. N.M. acknowledges the Boehringer Ingelheim Foundation Plus 3 Program, and the European Research Council (ERC-CoG, 724209). L.B.C. is funded by the Airforce Office of Scientific Research (FA9550-22-1-0207) and the National Institutes of Health (1DP2GM149549-01).

## Author contributions

C.F.K. and T.L. conceived the research project and designed the experiments. C.F.K., T.L., and X.C. performed experiments. T.L. and

C.F.K. analyzed the results. C.F.K. purified proteins. T.L. and C.F.K. wrote the manuscript. P.S., L.B.C., X.C. and L.B. helped improve the manuscript. P.S., N.M., C.F.K. and L.B.C. provided resources.

## Funding

## Competing interests
The authors declare no competing interests.
