## [Peer Review File · Nature Communications]

Membrane-induced 2D phase separation of the focal adhesion protein talinREVIEWER COMMENTS

Reviewer #1 (Remarks to the Author):

Litschel and Kelley, et. al. report that key components of focal adhesions, talin, vinculin, and pip2 lipids, form flexible condensates in solution and on membrane surfaces. As expected, PIP2 is required to recruit the proteins to the membrane, allowing them to form condensates. Optical trap experiments show that protein-protein interactions within condensates are strong, perhaps supporting the mechanical role of focal adhesions. The in vitro work is performed in a rigorous manner, for the most part. However, the analysis of the condensates reveals that they are in many cases more solid than liquid. Overall, in the absence of comparison of these data to phenomena in live cells, it is not very clear what the "condensate" description provides to advance understanding of focal adhesions, where the molecules studied here have long been known to assemble.

1. As a general question, is the concentration of crowding agent used in these studies representative of physiological crowding? Can the authors provide a justification for it?
2. Recovery of Talin/Vinculin droplets is very slow compared to most other condensate systems. Many droplets do not appear round. Further, the authors comment that the droplets have matured to a less liquid state. Is this really LLPS? Why is it being presented as LLPS vs. aggregation/precipitation?
3. The authors state, "Interestingly, droplets with Tn2, SUVs and β 1D were much more viscous than droplets in our earlier experiments, clumping together instead of forming spherical droplets." Why are these structures being called droplets if they do not behave like liquids? What is the definition of droplets? The authors should explain their rationale. With sufficient crowding, virtually any protein will aggregate. How do the authors know that these aggregates have physiological significance? I have similar concerns about the "droplets formed in the presence of lipid bilayers, for which the authors state, "These clusters...recover partially after photobleaching (Fig. 3E)." Again, should this phenomenon be described as physiologically relevant phase separation?
4. It is interesting that PIP2 is concentrated by condensates. But PIP2 is known to be rather immobile in supported bilayers, such that the energetic barrier to clustering of pips may be reduced. Authors should check whether pip also becomes clustered when condensates form in on free membrane surfaces, such as GUV membranes. Using GUV assays would also make it clear whether membrane-bound protein condensates are liquid-like, as the edges of condensates would be more free to round, owing to greater mobility of the underlying lipids.
5. In figure 3, the authors should label vinculin in slb experiments to see where it goes. Does it colocalize with pip and talin, consistent with the authors' model? Does it recover after photobleaching? The authors' model assumes these things are true but they have not been tested in figure 3.
6. The authors state: " We found Tn2 most often formed multiple tethers between the beads, and interactions were strong enough to frequently pull the membrane-coated bead out of its optical trap." What does it mean to say that Tn2 "formed multiple tethers between the beads"? How do you know this is the case? How many tethers and what are they connecting? Do you mean that multiple pip2 lipids interacted with Tn2? If so, how do you measure that? Or is it just inferred from the high forces? If so, it is a rather circular argument.
7. The authors point out that they are the first to perform force spectroscopy on full-length talin. What has been learned here? What are the differences between full-length talin and the previously examined domains of talin? This area seems little explored by the authors.
8. Mechanism in figure 4 is already known based on existing literature. That is to say, we already know that talin, vinculin, and pip2 cluster together in focal adhesions. It is not clear what the condensate description, especially in the context of a "condensate" that is more solid than liquid is adding here.
9. Minor comment: words like "excitingly" and "importantly" are used frequently in the results section. In the view of this referee, it is better to let the readers decide how to feel about the data. If it is exciting, explain why it is exciting, rather than simply labeling it as such.

Reviewer #2 (Remarks to the Author):

The article by Litschel et al. is an interesting work on phase separation of focal adhesion proteins. The authors performed a series of in vitro experiments in which they convincingly show liquid-liquid phase separation of talin and vinculin in solution and on supported lipid bilayers and propose a possible mechanism of focal adhesion clustering induced by PIP2 containing membranes. Moreover, they characterize rupture forces of talin from PIP2 membranes by optical tweezers. Overall, the experiments are convincing and scientifically sound and represent an interesting set of biophysical characterization of the interactions occurring between focal adhesion proteins. In my opinion, the main weakness of the study is the lack of evidence of such processes in more realistic models such as cell cultures, which limits the article reach and general interest.

Below are my suggestions to improve the paper:

- The authors should introduce and discuss what are the experimental evidence of phase separation of focal adhesion proteins in living cells. Some sentences in the paper induce the reader to think that it is an established issue that phase separation occur at focal adhesions in living cells, but this is not clearly explained in the introduction and in the discussion.

For example, in the introduction (page 2): "Consistent with these characteristics, recent studies demonstrate that focal adhesion adaptor proteins can undergo phase separation, suggesting that phase separation may contribute to integrin clustering, nascent focal adhesion assembly (12, 13), focal adhesion maturation (14), and focal adhesion disassembly (15).". In the discussion: "Focal adhesions show liquid-like properties in cells, and share many characteristics with confirmed biomolecular condensates". The authors should clearly distinguish between studies made on reconstituted systems in vitro from studies made on living cells, clearly support their statements with appropriate reference to previous studies, and better put these statements in the right context.

- The authors should introduce and cite previous studies on the interaction between talin and PIP2 containing membranes and the biological role of this interaction. This is particularly important since the proposed mechanism of focal adhesion assembly rely on this interaction.

- It is generally believed that talin binds to integrins and that the connection between the extracellular matrix and the cell cytoskeleton occurs through the talin-integrin linkage. In the discussion (page 6), the authors say "Our results suggest that PIP2-induced cooperative talin-membrane interactions are necessary for talin to resist the high mechanical forces within focal adhesions (up to 10 pN per molecule (47)), since the talin-integrin linkage is a low affinity interaction (30 – 500 μ M) (29)". However, there is evidence that the talin-integrin link can be reinforced by other means (see for example Bodescu et al. PNAS 2023). This work would be greatly improved if the authors would explore the phase separation of talin on membranes bound to the β 1 integrin cytoplasmic domain, maybe in the presence of stabilizing proteins such as kindlin.

- The manuscript lacks a detailed description of the experimental condition of the in vitro experiments. The methods section "confocal microscopy of biomolecular condensates" describes the coverslip preparation, but not the experimental procedure.

- In the optical tweezers experiments, it seems that there is only a qualitative evaluation of the bead coating with the lipid bilayer.

- Supplementary fig. S17. The authors should show a control experiment of FEC with talin but without PIP2, to exclude the presence of non-specific interaction of talin with the silica bead coated with the lipid bilayer.

- Supplementary fig. S17, legend. Panel D) and E) are not shown.

- Supplementary fig. S18. There is no legend for panel D.

Reviewer #3 (Remarks to the Author):

Talin Review

While the details of integrin activation by talin binding and the formation of supramolecular complexes that form the focal adhesions responsible for the control of cellular processes such as migration have been investigated for decades, this article provides some new insight into the lipid phase separations following the binding of talin and vinculin to anionic membrane surfaces. Novel is the use of single molecule measurements to follow the formation of lipid condensates. Given the broad interest in the area of signal transduction, the work is appropriate for a high profile journal

such as Nature Communications. However, there are serious historical referencing omissions, and extensive use of the euphoric adjectives such as "Excitingly ... ". The historical perspective needs to be included before publication. Here is a brief suggestion of the prior literature:

First Mark Ginsberg and Ken Taylor with colleagues were the first to show inside-out signaling via talin binding, including showing the conformational changes in the distal domain by EM: Ginsberg, Taylor et al. Journal Cell Biology (2010) 188, 157

The specificity of talin binding to PiP2 membranes is well known and a biophysical description was also described in detail before: Sligar et al. (2016) BBA 1858, 1833 and Biochemistry 55, 5038.

Indeed there was an entire GLU grant effort led by Rick Horowitz focusing on cell migration and focal adhesion dynamics yielding paper by Kim, Ye, Taylor, Dai, Ginsberg, etc. such as:

(2011) Ann. Rev. Cell Dev. Biol. 27, 321

(2011) Blood 119, 26 - 33

(2011) J. Thromb. Haemost. 1, 20-25.

(2015) J. Biol. Chem 290, 27168

I do not think that the nice work described in the manuscript by Litschel et al. should be published without placing in a historical perspective.

Reviewer #1 (Remarks to the Author):

Litschel and Kelley, et. al. report that key components of focal adhesions, talin, vinculin, and pip2 lipids, form flexible condensates in solution and on membrane surfaces. As expected, PIP2 is required to recruit the proteins to the membrane, allowing them to form condensates. Optical trap experiments show that protein-protein interactions within condensates are strong, perhaps supporting the mechanical role of focal adhesions. The in vitro work is performed in a rigorous manner, for the most part. However, the analysis of the condensates reveals that they are in many cases more solid than liquid. Overall, in the absence of comparison of these data to phenomena in live cells, it is not very clear what the “condensate” description provides to advance understanding of focal adhesions, where the molecules studied here have long been known to assemble.

We thank the reviewer for the comments and suggestions. The reviewer brings up an important point in this initial comment by asking how we think the ‘condensate description’ poses an advancement to the understanding of focal adhesions. We agree that our initial manuscript lacked a detailed discussion of the implications and significance of our findings, and have changed the manuscript accordingly. We now address this in the following sections of the manuscript, which we have rewritten, and - as we trust - improved: The last two sentences of our abstract, most of the introduction and parts of the discussion. We would also like to extend this discussion below, in this response letter, which will be published alongside the manuscript.

Liquid-liquid phase separation (LLPS) is a purely physical process distinct from liquid-to-solid transitions like aggregation in which proteins usually lose their functionality. It also differs drastically from other forms of spatial organization in cells such as polymerization or the formation of highly organized macromolecular assemblies. While LLPS can be based on a variety of molecular interactions, the commonality is both the liquid state of the emerging structures and the principles that govern their formation. Long before the discovery in cells, physicists have been studying these processes and describing many aspects of LLPS with mathematical models. One key property of biomolecular condensates is their dynamic nature and the reversibility of their formation. Further, it is important to note that these characteristics still apply to viscous liquids, which we will elaborate further in our answers to later points of reviewer 1.

The recent discovery of biomolecular condensates has revolutionized how we think about protein-protein interactions within cells, and the role of their dynamics in regulating cellular processes. Indeed, the activity and regulation of many proteins and genes can be directly linked to phase separation of specific proteins and nucleic acids in the cell. Mis-regulated phase-separation is now considered a hallmark of many diseases, including neurodegeneration, infectious disease, and cancer (Alberti and Dorman, Annual Review of Genetics, 2019, DOI: 10.1146/annurev-genet-112618-043527). Modulating condensates offers a completely novel approach to drug discovery, creating the possibility of targeting pathogenic molecules once deemed undruggable through traditional small molecule approaches (Mitrea et al., Nature Reviews Drug Discovery, 2022, DOI: 10.1038/s41573-022-00505-4). This is reflected in the many new companies founded on exactly that approach in the last few years - demonstrating the far-reaching influence the discovery of biomolecular condensates has already had on how we approach cell biology (Ortolano, *Biotech start-ups and condensate targeted drugs*, Drug Discovery News, 2022, Volume 18 Issue 4).

Focal adhesions have been well characterized regarding their molecular composition and the protein-protein interactions critical to their formation. However, their dynamics are central to their role in cellular processes, and cannot be ignored. Their ability to form rapidly, mature into stable structures, or disassemble when appropriate remains thus-far unexplained by traditional cell biology or biochemistry approaches to understanding complex formation. By applying the well-studied physical understanding of liquid-liquid phase separation to cellular assemblies, in this work, we think we offer a mechanistic explanation that can describe these defining properties that govern the formation and dynamics of focal adhesions. We think it is particularly exciting that our results allow us to hypothesize a new type of phase separation that, to our knowledge, has not yet been discussed: The formation of 2-dimensional condensates from proteins in solution.

1. As a general question, is the concentration of crowding agent used in these studies representative of physiological crowding? Can the authors provide a justification for it?

The concentration of crowding agent in this study is consistent with or lower than in many published *in vitro* reconstitution studies using purified proteins (André et al., 2020, DOI: 10.3390/ijms21165908). In general, including macromolecular crowder for *in vitro* experiments is predicted to better reflect the crowded cellular environment, compared to experiments lacking a crowding agent. In cells, there is a large variety of macromolecules, from microRNAs to truly large structures such as the cytoskeleton. It is difficult to quantify crowding in cells, as it depends on the size and properties of both the large molecules (crowder) and small molecules (e.g. reactants). Still, there have been rough estimations on how macromolecules affect the “effective volume” in cells. Estimates approximate that between 10% and 40% of the cytosolic volume is occupied by macromolecules, depending on the cell type and other factors (Zimmerman et al, 1991, DOI: DOI: 10.1016/0022-2836(91)90499-v; Cheung et al., 2013, DOI: 10.1002/cyto.a.22277; Theillet et al., 2013, DOI: 10.1021/cr400695p).

While a quantitative comparison between the *in vitro* systems and actual cytoplasm is challenging and beyond the scope of our research, we feel confident in saying that the amount of crowder we use here results in conditions that are less crowded than those in many *in vitro* reconstitution experiments where concentrations of 10% and more are commonly used (Maharana et al., 2018, *Science*, DOI: 10.1126/science.aar7366; Guo et al., 2019, *Science*, DOI: 10.1038/s41586-019-1464-0; Yang et al., 2020, *Cell*, DOI: 10.1016/j.cell.2020.03.046; Guillén-Boixet et al., 2020, *Cell*, DOI: 10.1016/j.cell.2020.03.049). We now include a sentence in the supplementary methods in which we compare the concentration used here to the literature.

It is worth noting that many of our experiments do not include any crowding agent. In Figure S5 we carried out a parameter sweep for crowding agent and protein concentration. At high concentrations of protein, we see phase separation even in the absence of crowder. In Figure S13, we show that talin phase separates with the β 1D peptide and SUVs in the absence of a crowder. Most importantly, all experiments on SLBs are performed entirely without crowding agent. We noticed that we did not point that out in the main text of our manuscript, and have now included this important detail in the first paragraph about phase separation on SLBs. We added the sentence: “*These clusters form in the absence of*

macromolecular crowding, but require a minimum amount of PIP₂ in the membrane to form". This phase separation in the absence of crowder can be explained by a lowering of the LLPS threshold by increasing concentration locally through membranes, which is a phenomenon that has been studied by others (Snead et al., Nature Cell Biology 24, 461-470 (2022)., Ditlev, Journal of Molecular Cell Biology 13, 319-324 (2021)., Beutel et al, Cell 179, 923-936.e911 (2019)., Gao et al., Molecular Cell 82, 1313-1328.e1318 (2022)).

2. Recovery of Talin/Vinculin droplets is very slow compared to most other condensate systems. Many droplets do not appear round. Further, the authors comment that the droplets have matured to a less liquid state. Is this really LLPS? Why is it being presented as LLPS vs. aggregation/precipitation?

Biomolecular condensates exhibit a wide range of material properties, including rapidly rearranging low-viscosity liquids and slowly rearranging high-viscosity liquids (Frey et al., 2007, DOI: 10.1016/j.cell.2007.06.024; Weber, 2017, DOI: 10.1016/j.cel.2017.03.003). These variations result from the molecules involved in their formation (Alshareedah et al., 2021, DOI: 10.1038/s41467-021-26733-7), as well as the time taken for droplets to form and the conditions of the system. The presence of and interactions with other molecules such as nucleic acids or membranes, whether specific or not, influences droplet characteristics, but whether these interactions make condensates more fluid or solid depends on context and molecular factors (Elbaum-Garfinkle et al., 2015, DOI: 10.1073/pnas.1504822112; Zhang et al, 2015, DOI: 10.1016/j.molcel.2015.09.017). Over time or due to specific mutations, droplets can become more solid in nature, which has implications for cellular function (Maharana et al., 2018, DOI: 10.1126/science.aar7366; Molliex et al., 2015, DOI: 10.1016/j.cell.2015.09.015; Patel et al., 2015, DOI: 10.1016/j.cell.2015.07.047). However, all of these different material states are still achieved through liquid-liquid phase separation, clearly defined by a phase diagram and a liquid-like state, which is readily observable in our system e.g. through the partial recovery, the rounded shape of the droplets (even after droplet fusion) and therefore also differs greatly from protein aggregation.

We extended the discussion and description of the viscosity of Tn-Vn condensates in the manuscript. A paragraph in the discussion section dedicated to this now reads:

"Biomolecular condensates have been identified in a range of physical states; varying greatly in terms of dynamics, stability, life-span, and function. The talin condensates we observe in solution become more viscous over time, and protein turnover varies depending on components and composition, which is in agreement with other LLPS experimental and theoretical literature (28, 58-61). Even the 3D droplets that form in combination with PIP₂ SUVs, the most viscous droplets we observe, have a characteristic rounded shape indicative of LLPS, which suggests that droplets were originally much less viscous, but aged over time. Similarly, the recovery of 2D membrane-associated Tn2 condensates on PIP₂ SLBs is fairly slow with a high immobile fraction, similar to what has been described as bioreactive gels (59). This would be consistent with talin's low diffusivity laterally along the membrane in focal adhesions (62). The mechanosensitive nature of focal adhesions may require this increased viscosity, in order to withstand forces applied through actomyosin contraction. A similar explanation has been offered for the gel-like state of the centrosomal condensates, which must withstand forces applied by the spindle during mitosis (59, 63)."

While the FRAP experiment in Figure 1 shows the bleaching of an entire droplet and hence demonstrates the exchange of protein between the droplet and the dilute phase, we now also include an example of a FRAP experiment with bleaching of a region within a condensate in the supplements (supplementary Figure S3 and Movie 2). Photobleaching a small region within a condensate is a better experiment to examine how quickly molecules diffuse and rearrange within the condensate. While this recovery is still somewhat slow compared to other in vitro biomolecular condensates, indicating a high viscosity, we see a higher FRAP recovery, which at the end of the recording is still increasing, consistent with a high viscosity liquid. We also include a video in the paper with a large field of view showing condensate fusion (Movie 1). While this movie is referenced in the manuscript of our original submission, unfortunately we forgot to include the movie in the submission. This movie shows approximately 500 droplet-merging events over the time course and a large field of view. Following each fusion event, the resulting larger droplet takes on a round shape, clearly indicating the liquid nature of the condensates.

Finally, we also demonstrate the specificity of talin droplet formation through the mutation or removal of residues known to be critical for specific protein-protein interactions. For example, the A50I mutation in vinculin, which prevents talin-vinculin interactions, does not trigger condensate formation. Removing the C-terminal dimerization domain of talin also prevents the formation of vinculin-talin condensates. Reducing the strength of talin-BID peptide interactions with the S339L mutation in the talin head domain has a clear inhibitory effect on the formation of talin-BID condensates. As all of these residues have been well characterized previously, this data strongly support that specific protein-protein interactions are responsible for the formation of talin-based condensates, and not nonspecific protein aggregation or precipitation.

3. The authors state, “Interestingly, droplets with Tn2, SUVs and β 1D were much more viscous than droplets in our earlier experiments, clumping together instead of forming spherical droplets.” Why are these structures being called droplets if they do not behave like liquids? What is the definition of droplets? The authors should explain their rationale. With sufficient crowding, virtually any protein will aggregate. How do the authors know that these aggregates have physiological significance? I have similar concerns about the “droplets formed in the presence of lipid bilayers, for which the authors state, “These clusters...recover partially after photobleaching (Fig. 3E).” Again, should this phenomenon be described as physiologically relevant phase separation?

We hope we already responded to some of these concerns with our reply to the previous point by the reviewer, where we discuss that the aspect of viscosity has been extensively studied by others in the field. The reviewer is correct that the condensates that form from PIP₂-SUVs and Tn2 are very viscous and fluorescence recovers incompletely. However, as discussed above, we can confidently say that these droplets form through liquid-liquid phase separation, e.g. obvious from the characteristic rounded droplet shape in all SUV experiments with only Tn2. Even the non-spherical condensates with additional β 1D (Figure S13) consist of spherical “subunits”. This leads us to believe that in this specific case with β 1D, initially, condensates are fully liquid and can relax into a spherical shape, but then age quickly and become rapidly more viscous. When these viscous droplets come into contact with each other, they

partially merge, resulting in this ‘bubbly’ structure. We now shortly address this in the manuscript, with the following sentence: “*Condensates with Tn2, SUVs and β 1D were much more viscous than droplets in our earlier experiments and appear as assemblies of spherical droplets that clump together (Fig. S13), indicative of rapid aging.*” We think that in these experiments, rapid aging and hardening can be explained by the specific mode of molecular interactions. To elaborate: in all other experiments in our work, condensates form from soluble proteins, which only have very few interaction sites per molecule. We believe that our dispersion of nanometer-sized SUVs is the closest state to a solution we can achieve with lipids. Nonetheless, this leads to slightly different results compared to traditional in vitro LLPS experiments. Importantly, SUVs have many binding sites for Tn2, which allows for many more interactions, leading to a higher degree of interconnectivity and thus more viscous condensates. We assume it takes some time to fully establish this dense network of interactions, and thus aging plays an important role.

We argue that the apparent viscosity of our PIP₂-associated condensates does not diminish the relevance of LLPS. Moreover, it could even hint to a physiological role. In their review, Woodruff, Hyman and Boke (2017, DOI: 10.1016/j.tibs.2017.11.005) focus on how condensates that show low levels of dynamic behavior could fulfill specific physiological roles in the cell. They argue that more viscous “gel”-like states could be necessary to withstand forces in the cell, particularly in condensates involved in the cytoskeleton. As focal adhesions are the anchor points of the actin cytoskeleton, this argument could make sense for the condensates we observe as well. More generally, the lateral interactions between focal adhesion proteins on the membrane in the liquid state could be a factor that enables focal adhesion to withstand strong forces, while still being able to quickly assemble and disassemble.

Lastly, we now also include images of spinodal decomposition of talin2 on membranes which we did not include in the original submission. We added the results as a new supplementary figure: Figure S21. Spinodal decomposition is a phase separation phenomena with a characteristic pattern (Cahn, 1965, DOI: 10.1063/1.1695731). Spinodal decomposition has been studied extensively in non-biological systems and has also been studied on surfaces (Ball & Essery, 1990, DOI: 10.1088/0953-8984/2/51/006; Jones et al., 1991, DOI: 10.1103/PhysRevLett.66.1326). Observation of spinodal decomposition provides further evidence that the underlying phenomena is liquid-liquid phase separation, and excludes other mechanisms such as aggregation.

4. It is interesting that PIP2 is concentrated by condensates. But PIP2 is known to be rather immobile in supported bilayers, such that the energetic barrier to clustering of pips may be reduced. Authors should check whether pip also becomes clustered when condensates form on free membrane surfaces, such as GUV membranes. Using GUV assays would also make it clear whether membrane-bound protein condensates are liquid-like, as the edges of condensates would be more free to round, owing to greater mobility of the underlying lipids.

We thank the reviewer for this comment and we agree that results would likely differ, as lipid and protein mobility in/on free-standing membranes is indeed different (i.e. more mobile) compared to SLBs. While we would also be curious to see how clusters behave on GUV membranes, we don’t think these experiments would add physiological relevance and might distract from the scope of this paper. We added

additional SLB experiments to the paper and discuss membrane fluidity and diffusion in more detail in the paper. In addition, we want to expand this discussion in the following paragraphs in this letter:

As the reviewer correctly points out, lipids in GUVs are more diffusive compared to SLBs (Pincet et al., 2016, DOI: 10.1371/journal.pone.0158457). Importantly however, cell membranes are less diffusive than either of these model membrane systems by up to two orders of magnitude (Kusumi et al., 2005, DOI: 10.1146/annurev.biophys.34.040204.144637). Related to this, we previously had found the behavior of bacterial membrane proteins in vivo to be closer to SLBs than to GUVs (Martos et al., 2013, DOI: 10.1111/1462-2920.12295).

With respect to the mobility of PIP₂ within SLBs, we agree with the reviewer that most research groups previously reported (relatively) immobile PIP₂ in SLBs. Forming fluid SLBs with PIP₂ can be difficult and require a different approach compared to SLBs formed with lipids of lower charge density. We found protonation of PIP₂ to be crucial before forming the SLB. The method is loosely based on a paper by Braunger et al. (2013, DOI: 10.1021/la402646k) and optimized by the group of Gijsje Koenderink, who was kind enough to send us their protocol.

Thus, the PIP₂ in our membranes is highly mobile. We show this in a FRAP experiment in our Figure S14. Further, as part of these revisions, we updated this figure with experimental data with better time resolution, as the previous plot was rather low in resolution. We further added a supplementary Figure (Figure S19) showing that PIP₂-bound talin without vinculin (i.e. without condensate formation) is diffusive on our membranes.

Inspired by the reviewer's comment, we have added yet another supplementary figure with FRAP data. In Figure S18 we show a FRAP experiment where we bleach a region within a membrane clusters. We see that even within these clusters, PIP₂ is highly mobile and that FRAP recovery is rapid (in fact, too fast to be quantifiable with the slower time resolution chosen here). Further, we notice that similar to FRAP experiments on liquid droplets in 3D, there are two different time scales. While exchange within the cluster is fast, the molecular exchange with the "dilute phase" outside the cluster is rather slow. Tn2 recovery is also slow and incomplete, similar to the more viscous 3D droplets in our experiments with PIP₂ SUVs. As mentioned above, this is not the case for PIP₂-bound talin without condensate formation (Figure S19).

5. In figure 3, the authors should label vinculin in slb experiments to see where it goes. Does it colocalize with pip and talin, consistent with the authors' model? Does it recover after photo-bleaching? The authors' model assumes these things are true but they have not been tested in figure 3.

We added a supplementary figure (Figure S16) with images of labeled vinculin that co-localizes in membrane condensates with Tn2 and PIP₂.

Unfortunately we found the SNAP tag with fluorescent label on vinculin to affect protein dynamics, which is why we include it in so few experiments. As the reviewer likely knows, fluorescent tags can alter

the properties of proteins. While this is often not a problem when low fractions of labeled proteins are used, we found vinculin to be affected by our fluorescent tag even at lower fractions. We did not observe such an effect with the other fluorescently labeled proteins and lipids we use in this study. However, the use of fluorescently labeled vinculin did result in the formation of fewer condensates. The condensates that did form were higher in viscosity compared to condensates formed with unlabeled vinculin, as evident by not only the fluorescence recovery of the labeled vinculin, but also that of labeled talin. Due to this artifact, it was not possible to measure the dynamics of vinculin within talin droplets. We apologize for not mentioning this previously in the manuscript, which we now do in the figure caption of the above mentioned figure and in the methods of our manuscript in the section about protein purification: *“We found that vinculin which is fluorescently labeled via SNAP tag affects membrane condensate formation and dynamics, hence we avoided fluorescently labeled vinculin in most experiments.”*

It might be worth mentioning a previous work of ours, where, unrelated to condensate formation, binding of vinculin to talin and the membrane is a main focus (Kelley et al. 2020, DOI: 10.7554/eLife.56110). In this study, we show that recruitment of vinculin to PIP₂-containing membranes is dependent on talin. We found that the C-terminal tag on vinculin did not alter the ability of vinculin to bind to talin or bundle actin filaments in vitro, and suggests that the C-terminal tag interferes more specifically with LLPS, which would be interesting to explore in future work.

6. The authors state: “ We found Tn2 most often formed multiple tethers between the beads, and interactions were strong enough to frequently pull the membrane-coated bead out of its optical trap.” What does it mean to say that Tn2 "formed multiple tethers between the beads"? How do you know this is the case? How many tethers and what are they connecting? Do you mean that multiple pip2 lipids interacted with Tn2? If so, how do you measure that? Or is it just inferred from the high forces? If so, it is a rather circular argument.

We thank the reviewer for asking this, as this section of the paper was not clear. Our previous text indeed made it sound like one talin molecule can bind to multiple lipids. However, we intended to say that one bead can bind to multiple talin molecules which then each bind to PIP₂. We rewrote this part, which now reads: *“Our force spectroscopy results suggest that usually multiple Tn2 molecules form tethers between the beads. Force curves show multiple, sequential rupture events (Fig. S24B). Often interactions were strong enough to pull the membrane-coated bead out of its optical trap.”* Importantly, Figure S24B relates directly to this statement. Our data suggests that the biotinylated bead interacted with multiple talin molecules (bound to the membrane-coated silica bead) at once. Four observations contribute to this conclusion, which we now more clearly point out in the manuscript and in the figure caption of Figure 18. (1) Multiple rupture events were observed (Figure S24B). In this case, after forming a connection between the two beads, large decreases in force are observed before complete rupture of the connection. (2) Additionally, the slope of the force curve is initially much steeper, and becomes less steep after each rupture event. These rupture events are much larger events than the unfolding events observed for an individual protein domain, as seen in Figure S24A for a single protein tether. (3) Very strong connections that do not rupture result in melting of the DNA tether (Figure S24C). In this case, a very steep force curve is observed, with noticeable unfolding events before DNA melting occurs. This indicates that the interactions between the (multiple) proteins and the membrane are able to withstand forces greater than

those necessary to unfold the DNA tether (>60 pN). After the DNA melts, multiple rupture events can be observed. (4) Finally, in many instances, a connection was detected between two beads, but measurements were impossible as the streptavidin coated-silica bead was pulled out of the optical trap due to overwhelming interactions with the polystyrene bead.

7. The authors point out that they are the first to perform force spectroscopy on full-length talin. What has been learned here? What are the differences between full-length talin and the previously examined domains of talin? This area seems little explored by the authors.

We agree that more context was needed to explain the force spectroscopy experiments with full-length talin. We added a section explaining this, and summarizing past results with the talin rod domains. We hope this clarifies the reasoning behind and significance of measuring these interactions and unfolding events in the context of the full-length, membrane bound talin. The following text has been added to the manuscript, and additional citations and conclusions have been added to other parts of the results section and the figure caption.

“To elucidate the interactions between talin and the membrane, we developed an assay to measure the strength of single Tn2-PIP₂ interactions using optical tweezers (Fig. 5A). Previous force spectroscopy experiments have shown that both talin-vinculin and talin-integrin interactions can be mediated or strengthened through force application to the talin rod domain (44-46). However, none of these experiments have explored the effect of force in the context of the full-length, autoinhibited talin protein. Talin’s N-terminal FERM domain secures the talin rod in a globular, inactive conformation in solution (22). We propose that upon binding to PIP₂ in the plasma membrane, the FERM domain releases the talin rod domain, making it available for force transduction.

To test whether the talin rod has the same mechanosensitive response to tension when bound to the membrane through the FERM domain, we used a dual trap optical tweezer set-up to pull on membrane-bound full-length talin.”

8. Mechanism in figure 4 is already known based on existing literature. That is to say, we already know that talin, vinculin, and pip2 cluster together in focal adhesions. It is not clear what the condensate description, especially in the context of a "condensate" that is more solid than liquid is adding here.

We hope with our responses and reactions to the reviewer’s initial comment and points 2 and 3, we have addressed the significance of the condensate description. If we understand the reviewer correctly, this is one of the main concerns of this point as well.

Our intention with Figure 4 is to summarize our findings and provide a visual representation of how we predict 2D condensates to form from proteins in solution - a new concept that might be hard to imagine for some readers. We think including an illustration will be valuable for reaching a broader audience.

However, we agree with the reviewer that we should not give the impression of wanting to take credit for discovering new components of focal adhesions. Therefore, we put more emphasis on the aspects that are relevant to our study. We now mention condensates within the figure itself and make the aspect of protein activation a central point of the figure, which we hope underlines the phase separation better. We also added two more sentences to the figure caption: *“In solution these molecules are not able to interact. Binding to the membrane triggers a cascade of protein activation and interactions that lead to liquid-liquid phase separation into 2D clusters.”*

9. Minor comment: words like “excitingly” and “importantly” are used frequently in the results section. In the view of this referee, it is better to let the readers decide how to feel about the data. If it is exciting, explain why it is exciting, rather than simply labeling it as such.

We agree with the reviewer. We have addressed these instances and also removed “Interestingly”, which we had used even more frequently.

Reviewer #2 (Remarks to the Author):

The article by Litschel et al. is an interesting work on phase separation of focal adhesion proteins. The authors performed a series of in vitro experiments in which they convincingly show liquid-liquid phase separation of talin and vinculin in solution and on supported lipid bilayers and propose a possible mechanism of focal adhesion clustering induced by PIP2 containing membranes. Moreover, they characterize rupture forces of talin from PIP2 membranes by optical tweezers. Overall, the experiments are convincing and scientifically sound and represent an interesting set of biophysical characterization of the interactions occurring between focal adhesion proteins. In my opinion, the main weakness of the study is the lack of evidence of such processes in more realistic models such as cell cultures, which limits the article reach and general interest.

Below are my suggestions to improve the paper:

- The authors should introduce and discuss what are the experimental evidence of phase separation of focal adhesion proteins in living cells. Some sentences in the paper induce the reader to think that it is an established issue that phase separation occur at focal adhesions in living cells, but this is not clearly explained in the introduction and in the discussion.

For example, in the introduction (page 2): “Consistent with these characteristics, recent studies demonstrate that focal adhesion adaptor proteins can undergo phase separation, suggesting that phase separation may contribute to integrin clustering, nascent focal adhesion assembly (12, 13), focal adhesion maturation (14), and focal adhesion disassembly (15).”. In the discussion: “Focal adhesions show liquid-like properties in cells, and share many characteristics with confirmed biomolecular condensates”. The authors should clearly distinguish between studies made on

reconstituted systems in vitro from studies made on living cells, clearly support their statements with appropriate reference to previous studies, and better put these statements in the right context.

We agree with the reviewer that this distinction is important. We have rewritten the section in our paper about existing literature that links focal adhesions to liquid-liquid phase separation. Further, while in the above quoted sentence all referenced publications are primarily in vivo, we've now extended our discussion of this existing literature with in vitro work and mathematical modeling. This new section now reads:

“Consistent with the above mentioned properties of focal adhesions, recent in vitro and cellular studies demonstrate that several focal adhesion adaptor proteins can undergo phase separation, and that phase separation may contribute to focal adhesion formation and function. Phase separation of the adaptor proteins FAK, paxillin, and p130Cas can promote kindlin-dependent integrin clustering on supported lipid bilayers and initial nascent focal focal adhesion assembly in cells (13-15). The mechanosensitive adaptor protein LIMD1 undergoes phase separation in vitro and in cells to form a condensed phase that recruits a specific subset of focal adhesion proteins. LIMD1 is recruited to focal adhesions under force, and perturbing LIMD1 phase separation alters focal adhesion dynamics, cellular mechanics, and durotaxis (16). The focal adhesion protein Tensin1 undergoes phase separation during focal adhesion disassembly (17). Thus, growing experimental evidence suggests that phase separation contributes to focal adhesion formation, maturation, and disassembly.”

- The authors should introduce and cite previous studies on the interaction between talin and PIP₂ containing membranes and the biological role of this interaction. This is particularly important since the proposed mechanism of focal adhesion assembly rely on this interaction.

We appreciate the suggestion, which we agree will make our manuscript better. We added a few sentences to introduce existing findings regarding the interaction between talin and PIP₂:

“[...] the phosphoinositide PIP₂ plays a major role in regulating talin localization and activation at the plasma membrane, and PIP₂ is necessary for proper formation of functional focal adhesions (39). When autoinhibited, talin's integrin-binding sites, as well as actin and vinculin binding sites within the rod domains, are obscured. The F2F3 domains of the talin head domain have a strong preference for PIP₂, even when compared to other acidic phospholipids (40, 41). Binding to PIP₂ is predicted to trigger a shift from a globular, inactive conformation to an open, extended conformation, revealing the integrin binding sites within the F3 domain. The talin rod domains would then be released from interactions with the talin head domain, and available to recruit vinculin and actin to membrane surfaces (22, 24, 42). Additionally, talin-membrane interactions are likely required to trigger the conformational change leading to integrin activation (32). Due to the close link between PIP₂ and talin engagement within FAs, we wished to test whether PIP₂-containing membranes can also trigger phase separation by releasing talin autoinhibition.”

- It is generally believed that talin binds to integrins and that the connection between the extracellular matrix and the cell cytoskeleton occurs through the talin-integrin linkage. In the discussion (page 6), the authors say “Our results suggest that PIP₂-induced cooperative talin-

membrane interactions are necessary for talin to resist the high mechanical forces within focal adhesions (up to 10 pN per molecule (47)), since the talin-integrin linkage is a low affinity interaction (30 – 500 μ M) (29).”. However, there is evidence that the talin-integrin link can be reinforced by other means (see for example Bodescu et al. PNAS 2023). This work would be greatly improved if the authors would explore the phase separation of talin on membranes bound to the β 1 integrin cytoplasmic domain, maybe in the presence of stabilizing proteins such as kindlin.

Thank you for drawing our attention to this recent and highly relevant study about the role of kindlin in stabilizing the talin-integrin interaction. We now briefly mention kindlin in the introduction (see our response to the first point by reviewer 2), and also discuss the potential contribution of kindlin to enabling high force transmission in the discussion:

“Additionally, kindlin may further cooperate with talin and PIP₂ to enable transmission of high mechanical force at focal adhesions (Bodescu PNAS, 2023).”

- The manuscript lacks a detailed description of the experimental condition of the in vitro experiments. The methods section “confocal microscopy of biomolecular condensates” describes the coverslip preparation, but not the experimental procedure.

The reviewer is absolutely right and we apologize for the previously bad state of our methods section. We indeed did not include any information about the preparation of the actual experimental sample and the handling of the protein immediately before microscopy. We added a new paragraph in the section about (3D) biomolecular droplets where we protocol the previously missing information about experimental conditions for these experiments. We also revisited other parts of our methods section and hope we overall improved the methods section of the paper.

- In the optical tweezers experiments, it seems that there is only a qualitative evaluation of the bead coating with the lipid bilayer.

Unfortunately we’re not sure what the reviewer is implying here, e.g. what a possible quantitative evaluation would look like. In our tweezer experiments we always had a visual indication regarding the membrane-coating, as we used fluorescently labeled lipids and thus were able to confirm that a membrane had formed around the beads. We agree that our work was missing some sort of confirmation that PIP₂ was present on our beads, which we now include, also as part of our response to the reviewers next point below.

- Supplementary fig. S17. The authors should show a control experiment of FEC with talin but without PIP₂, to exclude the presence of non-specific interaction of talin with the silica bead coated with the lipid bilayer.

We performed the experiment the reviewer has asked for, which is now panel D in supplementary Figure S22 (formerly Figure S17).

- Supplementary fig. S17, legend. Panel D) and E) are not shown.

We apologize for the negligence. The panels that were described in the figure caption are now added as a separate figure (Figure S23).

- Supplementary fig. S18. There is no legend for panel D.

We have added the figure panel legend to the figure, which is Figure S24 in the updated version of the manuscript.

Reviewer #3 (Remarks to the Author):

Talin Review

While the details of integrin activation by talin binding and the formation of supramolecular complexes that form the focal adhesions responsible for the control of cellular processes such as migration have been investigated for decades, this article provides some new insight into the lipid phase separations following the binding of talin and vinculin to anionic membrane surfaces. Novel is the use of single molecule measurements to follow the formation of lipid condensates. Given the broad interest in the area of signal transduction, the work is appropriate for a high profile journal such as Nature Communications. However, there are serious historical referencing omissions, and extensive use of the euphoric adjectives such as “Excitingly ... “. The historical perspective needs to be included before publication. Here is a brief suggestion of the prior literature:

First Mark Ginsberg and Ken Taylor with colleagues were the first to show inside-out signaling via talin binding, including showing the conformational changes in the distal domain by EM: Ginsberg, Taylor et al. Journal Cell Biology (2010) 188, 157

The specificity of talin binding to PiP2 membranes is well known and a biophysical description was also described in detail before: Sligar et al. (2016) BBA 1858, 1833 and Biochemistry 55, 5038.

Indeed there was an entire GLU grant effort led by Rick Horowitz focusing on cell migration and focal adhesion dynamics yielding paper by Kim, Ye, Taylor, Dai, Ginsberg, etc. such as:

(2011) Ann. Rev. Cell Dev. Biol. 27, 321

(2011) Blood 119, 26 - 33

(2011) J. Thromb. Haemost. 1, 20-25.

(2015) J. Biol. Chem 290, 27168

I do not think that the nice work described in the manuscript by Litschel et al. should be published without placing in a historical perspective.

We agree with the reviewer that we had not sufficiently covered previous studies in our discussion. Particularly of interactions between talin and integrin, but also between talin and PIP₂, as the reviewer has pointed out. We now address this in two new sections in the manuscript in which we cite all 6 publications the reviewer mentioned:

“Integrin receptors are the foundation of focal adhesions, connecting mechanosensitive and signaling machinery within the cell to ligands in the extracellular environment. Integrins are made up of non-covalent heterodimers, referred to as the α - and β -subunits, which switch between a bent, inactive form and an extended, engaged, active conformation when bound to an extracellular ligand (29). The α -integrin cytoplasmic domain binds directly to the talin head via two distinct interaction sites in the F3 domain (30, 31). This interaction is critical for integrin activation and signaling, as talin binding triggers a conformational change separating the α - and β -subunits, thereby increasing integrin’s affinity for extracellular ligands (32-34).”

and

“When autoinhibited, talin’s integrin-binding sites, as well as actin and vinculin binding sites within the rod domains, are obscured. The F2F3 domains of the talin head domain have a strong preference for PIP₂, even when compared to other acidic phospholipids (40, 41). Binding to PIP₂ is predicted to trigger a shift from a globular, inactive conformation to an open, extended conformation, revealing the integrin binding sites within the F3 domain. The talin rod domains would then be released from interactions with the talin head domain, and available to recruit vinculin and actin to membrane surfaces (22, 24, 42). Additionally, talin-membrane interactions are likely required to trigger the conformational change leading to integrin activation (32).”

REVIEWER COMMENTS

Reviewer #1 (Remarks to the Author):

The authors have made a thoughtful attempt to address concerns expressed about their original submission. Most aspects have now been clarified. A few remain.

1. Near the beginning of the paper (abstract/introduction), the authors make claims that sound as if they are the first to find proteins that phase separate out of solution onto membrane surfaces in order to achieve a biochemical function. This is simply not the case, as reports similar in concept have come from many groups over the past 10 years (Vale, Groves, Rosen, Stachowiak, Gladfelter, etc.) in many different systems (cytoskeleton, immunological synapse, endocytosis, control of transcription, etc.) The text should be thoroughly clarified in this aspect.

2. The authors seem to argue that no matter how viscous a "condensate" becomes, it can still be called a condensate rather than a "solid" or "aggregate". It seems to me that by this definition, all forms of condensed matter are "condensates". If so, then I fail to see the novelty of the present findings, as the proteins and lipids under study here have long been known to "condense" at the membrane surface. If the authors are arguing for a more nuanced definition of a "condensate", then they must define it. In particular, the reader needs to understand which protein assemblies are NOT condensates. For example, are well-characterized structured protein assemblies (actin, microtubules, clathrin lattices) included in the condensate definition? How does one measure or test a material to decide if it is a "condensate"? At a minimum the authors should define at least one example of a protein assembly that is NOT a condensate and show that it behaves in a way that is physically distinct according to one or more physical tests. This is particularly important when dealing with protein assemblies that are often near the diffraction limit in at least one dimension. In such cases, many structures will appear "rounded" owing to diffraction and recovery after photo-bleaching, albeit at slow rates, may occur even in "solid" materials, where molecules still have some limited mobility.

Reviewer #2 (Remarks to the Author):

The authors have significantly improved the manuscript by describing in more detail the existing literature and the experimental methods used. Unfortunately, they did not explore the phase separation of talin on membranes bound to the $\beta 1$ integrin cytoplasmic domain as I suggested in my previous comments, which in my opinion limits the reach of the article. However, I believe that the article is scientifically sound and now suitable for publication in Nature Communications. As a minor comment, panel D in figure S22 lacks the caption and lacks discussion of the experimental results.

Reviewer #3 (Remarks to the Author):

This is a much improved manuscript and seems to address all of the reviewer comments. It is of significant interest to warrant publication in Nature Communications

1. Near the beginning of the paper (abstract/introduction), the authors make claims that sound as if they are the first to find proteins that phase separate out of solution onto membrane surfaces in order to achieve a biochemical function. This is simply not the case, as reports similar in concept have come from many groups over the past 10 years (Vale, Groves, Rosen, Stachowiak, Gladfelter, etc.) in many different systems (cytoskeleton, immunological synapse, endocytosis, control of transcription, etc.) The text should be thoroughly clarified in this aspect.

We did not intend to disregard pioneering work by other groups and apologize if our manuscript has come across like this. We have revised the introduction of our manuscript and now try to highlight more precisely what we think the novel aspects of our work are, while also mentioning the work brought up by the reviewer. As part of this process we have reorganized large parts of the introduction and think we overall improved our manuscript significantly.

2. The authors seem to argue that no matter how viscous a "condensate" becomes, it can still be called a condensate rather than a "solid" or "aggregate". It seems to me that by this definition, all forms of condensed matter are "condensates". If so, then I fail to see the novelty of the present findings, as the proteins and lipids under study here have long been known to "condense" at the membrane surface. If the authors are arguing for a more nuanced definition of a "condensate", then they must define it. In particular, the reader needs to understand which protein assemblies are NOT condensates. For example, are well-characterized structured protein assemblies (actin, microtubules, clathrin lattices) included in the condensate definition? How does one measure or test a material to decide if it is a "condensate"? At a minimum the authors should define at least one example of a protein assembly that is NOT a condensate and show that it behaves in a way that is physically distinct according to one or more physical tests. This is particularly important when dealing with protein assemblies that are often near the diffraction limit in at least one dimension. In such cases, many structures will appear "rounded" owing to diffraction and recovery after photo-bleaching, albeit at slow rates, may occur even in "solid" materials, where molecules still have some limited mobility.

We regret that in our first response letter, we evidently did not address all of the reviewers' concerns, and are happy to clarify things further. We have modified a few parts of our manuscript, but would also like to add a detailed response here. In the following, we think we have addressed all concerns the reviewer has raised:

We would like to start by answering the reviewers first point regarding the placement of LLPS condensates in condensed matter physics. Liquid is a clearly defined physical state of matter and liquid-liquid phase separation is a clearly defined physical concept. Liquids are fluid, which means, they can flow if a shear force is exerted onto them. This force can be surface tension leading to fusion of liquid droplets, and can result in the relaxation of droplets into circular or spherical shapes. Liquids lack internal order, therefore are anisotropic, amorphous structures and exhibit complete rotational (and translational) symmetry. Molecules in a liquid diffuse, which can be observed on a confocal microscope, when a region within a liquid is photobleached. If a region within a solid is photobleached, the signal in this region will not recover due to diffusion. Solids also cannot fuse and relax back into a rounded or spherical shape. This means, with relatively straightforward methods, solids can be distinguished from liquids. From a mechanical perspective, there are clear differences between solids and liquids in their response to shear stresses: Solids

are elastic, liquids are viscous. Solids have elastic properties, but do not have a viscosity. This means that solids respond with a restoring force to shear stresses, which liquids do not. Instead of elasticity, viscosity is the defining property of liquids, which describes a resistance to flow. Even the most viscous liquid does not have these elastic properties. Liquids can have extreme levels of viscosity, as, for example, demonstrated in the famous pitch drop experiments, but these highly viscous liquids still lack any of the clearly defined physical properties of solids.

Actin and microtubules are a great example of structures that exhibit none of these defining properties of liquids. They are not amorphous, are very ordered, and have a repeating structure and no rotational symmetry, except about one axis. There is long range order and correlation in biopolymers like actin. Polymers have measurable properties like persistence lengths, bending rigidity and other properties that can be determined for polymers and solids – properties that are absent in liquid condensates. All of these properties have been studied by biophysicists and they greatly contributed to our understanding of these structures. Experimentally, actin filaments and other cytoskeletal filaments can easily be identified as solids, as these filaments do not recover after photobleaching and cytoskeletal filaments do not relax into a spherical shape when deformed. Slightly harder to distinguish from condensates are structures like aggregates, which are usually associated with misfolded or damaged proteins. However, aggregation in cells is a generally irreversible process and results in the formation of a gel, i.e. a solid structure. Subunits within these solid structures do not diffuse or flow. While aggregates can form as spherical objects, they do not relax back into spherical shapes after fusion and can be irreversibly deformed. Besides unwanted aggregation, there are few other amorphous structures in cells; historically these were often termed “granules”, many of which have meanwhile been identified as biomolecular condensates. Most of the remaining spatial organization of cells is achieved via highly ordered structures like membranes and polymers, making biomolecular condensates an exception and thus LLPS such an important concept.

Importantly, the laws that govern their formation are very different for biomolecular condensates and other structures like aggregates. This formation depends differently on factors like concentration, pH, salt concentrations or temperature. In the case of LLPS all of these parameters are used to describe a LLPS phase diagram, which explains precisely whether phases will separate, and if so, what the ratio between these two separated phase is and the concentration within the two phases (DOI: 10.1016/j.cell.2018.12.035). Actin and microtubules dynamics have been studied for decades and many groups have dedicated their research on the polymerization and depolymerization of these biopolymers and how other proteins are involved in this process. Aggregation, similar to polymerization, is a kinetic process that can be described with rate constants. These also depend on factors like concentration and temperature, but in a different way. Phase separation usually happens on short timescales, which is why it is often seen as a very dynamic process that allows rapid assembly and disassembly of these cellular structures.

The structures we show in our experiments are clearly liquid. We would like to recapitulate the evidence we have for the liquid nature of our 3-dimensional droplets, which seems to be the reviewer's original main concern. Many of the condensates we see in our experiments have a low viscosity, completely relax after fusion and become spherical and recover after photobleaching (although incompletely). All of which are properties that can only be found in fluids. Both movies in our manuscript show cases of condensates that should be easily identifiable as liquids: Movie 1 shows a FRAP experiment with clear recovery and intensity gradients (indicating diffusion) and Movie 2 shows many circular droplets and about 500 fusion events

followed by relaxation into larger, circular droplets. In other experiments FRAP recovery may be reduced and droplets restore their spherical shape incompletely after fusion, however, both of which are still evidence of a liquid state and solely indicate a higher viscosity. Further, we have no reason to assume that only some of the structures we show form through LLPS, but not others.

The reviewer is correct that in some cases, tests like FRAP or observing fusion can be misleading. Small solid structures like aggregates can appear to recover because of growth or exchange on the surface, and small puncta can attach to each other, and thus appear to merge and to result in a spherical structure. As the reviewer points out, this can be problematic if these structures are close to or below the diffraction limit of the microscope, as it is often the case when looking at (apparent) condensates in cells. However the condensates in our images are far from the diffraction limit. Many of our condensates, including in the above-mentioned movies, are between 5 and 50 μm in size. Bleaching within the volume of the condensates precludes any surface effects and clearly shows that the inside of our condensates is liquid. Our resolution is high enough that we can observe gradients and diffusion within the condensates. We now highlight this in our figure captions, which we had not previously included. All of the above clearly indicates a liquid phase.

Importantly, although viscosity of our condensates increases over time, this does not make them any less liquid, as viscosity is an inherent property of fluids (in fact, only of fluids - see above) and generally condensates are thought to remain fluid even after what is often mistakenly assumed to be gelation. As we discussed in our first response letter, many biomolecular condensates age over time, and become more viscous. This behavior has been described and discussed by many groups (see references in our first letter), and a great study regarding this was published by Jawerth et al. (Science, 2020). Often biologists describe this aging of condensates as “gelation” or “hardening” which is in fact a misleading term, as it implies a transition to a solid state. Jawerth et al. find that even viscous, aged condensates show the properties of a real fluid, while only the time dependent aging process in and of itself resembles that of physical glasses. Even if we assume our condensates become solid-like over time (which we do not have any reason to assume), the structures we show clearly form through liquid-liquid phase separation, and we think it is reasonable to say that cell biologists care about understanding the formation of cellular structures just as much as the properties in their final state.

While we think all of our experiments clearly show a liquid nature of our 3-dimensional condensates and that these form through liquid-liquid phase separation, the tests that we describe above, like FRAP, are less obvious regarding our 2-dimensional condensates. The reviewer mentions that sometimes FRAP recovery can be caused by factors other than diffusion within liquids. While this is not the case when a region within a 3D droplet is bleached, one could imagine the recovery of 2D droplets not to occur through diffusion within the condensate, but through exchange with the solution above. However, we think our data clearly support the conclusion that the 2D condensates form through the same interactions as the 3-dimensional droplets. We show that (i) talin undergoes LLPS in 3D, (ii) talin needs an interaction partner to do so and (iii) the phospholipid PIP_2 can be such an interaction partner: in experiments with PIP_2 within SUVs in 3D we see the formation of 3D structures that we can clearly identify as biomolecular condensates. This should be sufficient to conclude that the structures we see when PIP_2 is restricted to a 2D surface form through the same mechanism, i.e. liquid-liquid phase separation. In our supplements we provide further evidence that these are phase separated structures. In Figure S18 we see gradients of talin on the membrane after

bleaching, indicating that talin on the membrane diffuses and is in a liquid state. Finally, we also see spinodal decomposition (Figure S21) of talin on SLBs, which is a very direct indication of liquid-liquid phase separation.

We think all of the above should provide convincing evidence of the liquid nature of our droplets and that these proteins undergo LLPS and importantly, also why it matters for cell biology.

REVIEWERS' COMMENTS

Reviewer #1 (Remarks to the Author):

I have no further comments for the authors.